# An analysis of the accuracy of retrospective birth location recall using sibling data

Stephanie von Hinke [1,2,3] ✉ & Nicolai Vitt [1] ✉

Many surveys ask participants to retrospectively record their location of birth. This paper examines the accuracy of such data in the UK Biobank using a sample of full siblings. Comparison of reported birth locations for siblings with different age gaps allows us to estimate the probabilities of household moves and of misreported birth locations. Our first contribution is to show that there are inaccuracies in retrospective birth location data, showing a sizeable probability of misreporting, with 28% of birth coordinates, 16% of local districts and 6% of counties of birth being incorrectly reported. Our second contribution is to show that such error can lead to substantial attenuation bias when investigating the impacts of location-based exposures, especially when there is little spatial correlation and limited time variation in the exposure variable. Sibling fixed effect models are shown to be particularly vulnerable to the attenuation bias. Our third contribution is to highlight possible solutions to the attenuation bias and sensitivity analyses to the reporting error.

Retrospective data collection on residential locations is common in secondary data sources. For example, many datasets include individuals' residential location at birth or in early childhood, recollected by survey participants in adulthood or older age (e.g. the US National Longitudinal Survey of Youth 1979, the UK Biobank (UKB) and Understanding Society, the German Socio-Economic Panel, the Dutch Lifelines cohort and the Generation Scotland cohort). These location data have in turn been used in a wide range of empirical applications, such as those studying geographic mobility[1,2], geographic stratification and spatial correlation of genetic variation[3–8], assortative mating and social homogamy[9,10], but they have also been used to capture regional differences in infrastructure, health or economic circumstances, such as the staggered roll-out of policy[11]. Similarly, they have allowed researchers to include area of birth fixed effects to account for systematic differences between geographical areas[12,13], and to merge in external information on (area-level) weather, health or socio-economic information[14–21]. We discuss a number of high-profile studies that use the birth location data in different empirical applications in Supplementary Note 1. Despite much research in a wide range of applications relying on these retrospectively recorded (birth) locations, very little work has explored the accuracy of these data, especially considering they often rely on individuals' correct 30+ year recall. With that, there

is also little work on the consequences of such inaccuracies, as well as how these depend on different (spatial and temporal) parameters of the data-generating process. This is the aim of our paper.

We focus on the UKB, a large cohort study of half a million individuals aged 45–69 between 2006 and 2010. It includes detailed environmental, lifestyle, health and genetic data, but has very limited information on the environment in which individuals grew up and the circumstances during their early childhood. It does, however, record individuals' location of birth. These data are based on the following question which was asked by the interviewers to any participant who indicated being born in England, Scotland or Wales: "What is the town or district you first lived in when you were born?" Based on the respondent's answer, the interviewer selected the corresponding place from a very long and detailed list of place names in the UK. The birthplace was then converted into coordinates (eastings and northings at a 1-km resolution) which are provided in the data. Supplementary Background 1 illustrates how birth locations are mapped to 1-km coordinates.

The first contribution of this paper is to explore the accuracy of these data. We exploit the fact that the UKB includes a sample of approximately 40,000 full siblings which can be identified using the genetic kinship matrix. We start by constructing a binary variable

[1]School of Economics, University of Bristol, Bristol, United Kingdom. [2]Institute for Fiscal Studies, London, United Kingdom. [3]Institute for the Study of Labor (IZA), Bonn, Germany. ✉e-mail: s.vonhinke@bristol.ac.uk; nicolai.vitt@bristol.ac.uk

indicating whether two siblings reported different locations of birth. Assuming that the siblings grew up together, their location of birth can differ for two reasons. First, the family may have moved house between the births of their two or more children (i.e. a 'true' change in their birth location). Second, there may be an error in the birth location recorded for (at least) one of the siblings. For the former, we assume that a longer spacing between births linearly increases the probability of a house move; something we test empirically below. The latter can occur due to three reasons. First, a UKB participant could have incorrectly recalled their location of birth. Second, any differential recording across interviewers can cause errors in the location (e.g. recorded at different levels of detail). Third, it may have been caused by errors in processing the reported birthplace to grid coordinates if siblings differ in the precision of their reporting (holding constant the interviewer). We refer to the latter three 'location errors' as measurement error.

We examine the relationship between the differences in siblings' reported birth location and the age gap between the siblings. This allows us to derive the probabilities of house moves as well as misreporting. We explore heterogeneity in these probabilities across a wide range of factors, including birth cohorts, district types, population density, UKB assessment centre locations (at which the birth location was recorded), region of birth, siblings' sex, districts' socio-economic composition and siblings' polygenic index (PGI) for education. We explore the robustness of our error probability estimates using a subsample of UKB participants who reported their birthplace twice, ruling out house moves and reducing any individual reporting error.

The second contribution of this paper is to highlight the potential implications of using retrospective birth location data for research. It is well-known that conventional measurement error in explanatory variables causes attenuation bias in the estimated coefficients in a linear regression. To the extent that individuals' recorded birth locations reflect conventional measurement error, using birth locations (or external data merged based on birth locations) as explanatory variables will lead to attenuation bias. For many applications, however, the measurement error is unlikely to be classical. Allowing for non-classical measurement error, we explore the extent of this attenuation bias using Monte Carlo simulations for measures of disease exposure, demographic variables, and simulated spatial data with varying levels of spatial correlation and time variation.

The third contribution is to discuss potential solutions to the attenuation bias and possible sensitivity analyses to the reporting error. Although any solution will depend on the empirical analysis and the sample of interest, we discuss three possibilities, highlighting their advantages as well as potential drawbacks.

## Results
### Differences in siblings' birth location
We start by graphically presenting the unadjusted relationship between the discordance of siblings' birth locations and their age gap in Fig. 1. Panels a–i show the relationship for different levels of birth location accuracy. Panels a–c plot the share of sibling pairs reporting different parishes, districts or counties of birth, respectively. This shows that 28–30% of twins (i.e. siblings with an age gap of 0 years) report coordinates in different parishes and districts, with 8% reporting birth coordinates that are located in different counties. Furthermore, the graphs show a clear increase in this discordance as the age gap between siblings increases. This is expected since an increase in birth spacing also increases the likelihood of a house move between the two births.

To explore what may be driving the relatively large proportion reporting a different geographic area of birth for those born within a small age gap, we examine the shares of siblings reporting birth locations more than 0, 5, 10, 20, 30 and 50 km apart in panels d–i. This shows a similar positive relationship between the age gap and the probability of siblings reporting different coordinates. In fact, we show below that the slope coefficient in a linear regression is very similar across specifications. Furthermore, we show that the discordance between siblings is mainly driven by relatively small differences in eastings and northings. Indeed, 42% of twins report birth location coordinates that differ (i.e. are more than 0 km apart), but this reduces to 21% when we define discordance as those who report locations at least 5 km apart, 11% at 10 km and 6% at 20 km. From 30 km onwards, the discordance share among twins is fairly stable at 3%.

We next quantify the relationship between the siblings' age gap and the discordance of their reported birth location further using a linear regression. In addition to the estimated discordance for twins (i.e. those with an age gap of zero; the constant), the top panel of Table 1 shows that each additional year between the birth of two siblings increases the probability of reporting different coordinates by approximately 1 percentage point. While this estimate is very similar across the different specifications, it does decrease with distance, indicating that the probability of a long-distance move is lower than the probability of any move.

Finally, we investigate potential non-linearities in the relationship between the discordance of birth location and the siblings' age gap. Supplementary Table 1 shows that adding a quadratic term only marginally changes the estimated probabilities, suggesting that the linear specification in Table 1 is appropriate.

### Derived probabilities of household moves and measurement error
The regression estimates in the top panel of Table 1 can be used to derive the estimated annual probabilities of a household move, which we define as $\hat{q}$ (equivalent to the slope coefficient on the sibling age gap in the top panel), as well as the probability of measurement error in the reporting of an individual's birth locations, denoted by $\hat{p}$ (derived using Equation (4) in the 'Methods' section). These derived probabilities are shown in the bottom panel of Table 1. These derivations rely on a number of simplifying assumptions: (1) if siblings report the same birth location, we assume this is the true birth location, (2) biological siblings have grown up in the same household, (3) the move probability increases linearly with the age gap between the two siblings and (4) errors in the birth location occur randomly with the same probability across all participants, are independent within sibling pairs and independent of the sibling age gap. A more detailed discussion of these assumptions can be found in the 'Methods' section.

The probability of an error in participants' birth coordinates (at a 1-km resolution; column 4) is estimated to be 28.4%. However, a large share of these errors is due to small differences in the birth location with the estimated error probability reducing to 13.7% and 8.3% for differences of more than 5 and 10 km, respectively. Similarly, the estimated probability of an error in a participant's birth parish and district are 16.8% and 15.8%, respectively. Errors with a large difference in birth location are relatively rare, with an estimated error probability of 6.3% for participants' county of birth, and 3.4% for birth location differences of more than 50 km.

In Supplementary Results 2 we explore heterogeneities in the annual probability of a household move and the probability of measurement error along several dimensions. We find that later (i.e. younger) cohorts and those living in rural, less densely populated areas exhibit more measurement error. Our results furthermore show that mobility is higher among families initially living in rural and highly educated areas. Finally, we observe substantial differences in measurement error across the assessment centre locations at which participants completed their initial interview, as well as individuals' regions of birth.

In Supplementary Results 3 we conduct a supplementary analysis of the measurement error in birth location using a UKB sub-sample of participants who were asked to report their birth location more than

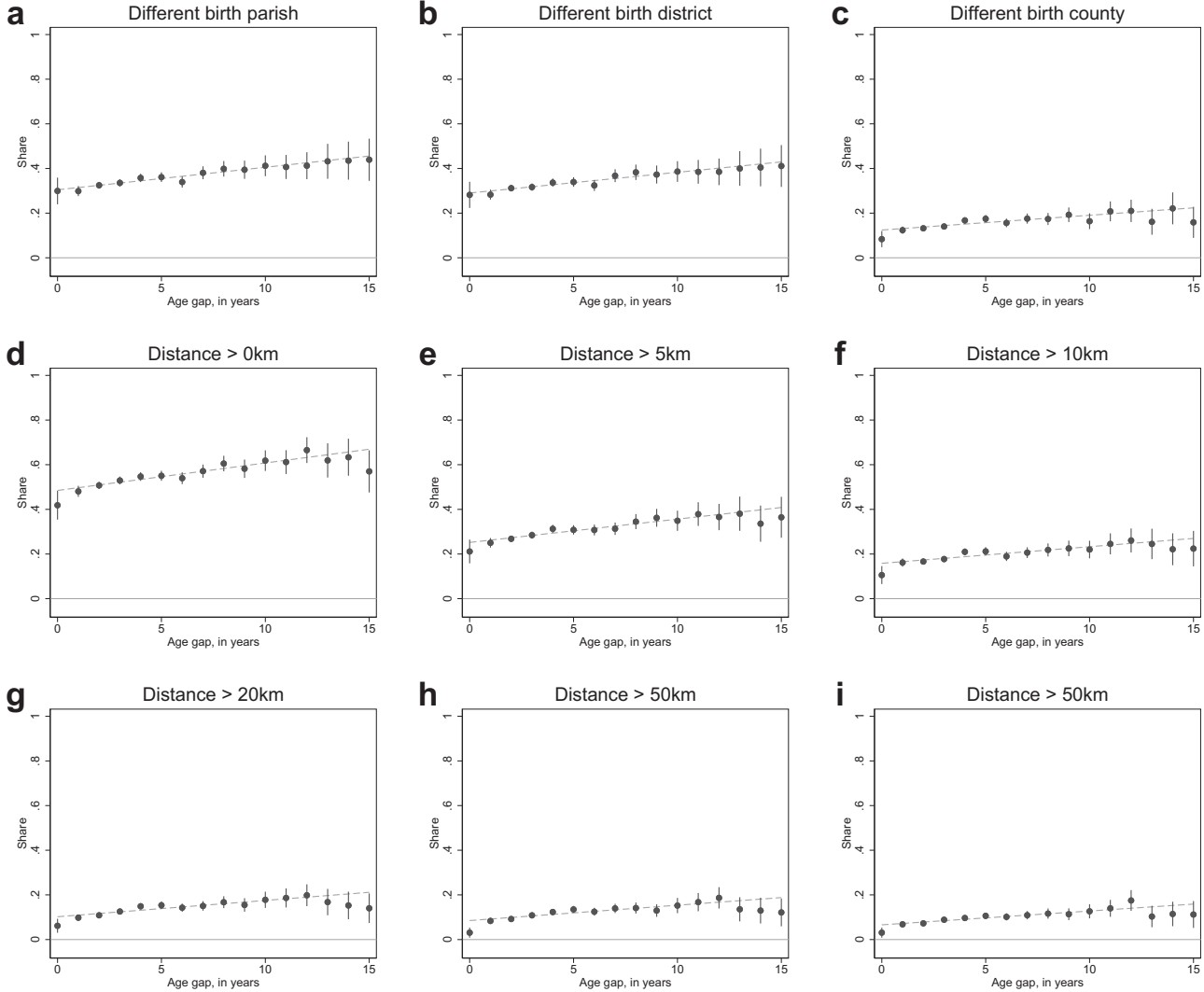

**Fig. 1 | Differences in siblings' birth location and their age gap. a–i** Birth location differences at different levels of location accuracy as indicated above each panel. Points are the share of sibling pairs reporting different birth locations among sibling pairs with the given (rounded) age gap. Vertical bars represent 95% confidence intervals. The dashed line is a linear fit for the relationship between the discordance share and the age gap. Source data are provided as a source data file. The graph is based on a sample size of $n = 18,314$ sibling pairs. Due to the small number of sibling pairs with an age gap of 15.5 years or above (165 pairs), these are omitted from this graph.

once. This allows us to distinguish between variation driven by house moves (which can be ruled out in repeated reports of birth location) and measurement error (which cannot). Similar to the sibling approach, the error probability estimated in this sub-sample captures interviewer effects, since it is likely that interviewers differed between siblings and between repeat interviews. Additionally, the repeat measures will also capture time-varying individual-reporting errors, which in turn may also lead to processing errors. Time-invariant individual reporting errors, however, will not be captured in the analysis of repeat measures.

Our findings show that the error probabilities in participants' birth locations that are based on the repeat measures are lower than those based on the sibling comparison in Table 1. For small error distances up to 5 km as well as for parish and district errors, the difference in probabilities to the sibling comparison is relatively small (e.g. 24.6% vs 28.4% for any error distance). However, for larger error distances the error probabilities derived from the repeat location reports are substantially lower (e.g. 1.2% vs 3.4% for error distances over 50 km). Thus, our results suggest that short-distance errors are mostly driven by the interviewer, time-varying individual reporting and processing effects

(which are captured to a similar degree by both approaches). Long-distance errors, on the other hand, are more likely to be driven by time-invariant individual reporting errors (which are only captured in the sibling comparisons).

**Attenuation bias**

Measurement error in birth locations may lead to attenuation bias when investigating the impact of birth location-based variation in exposures. The bias is increasing in the variance of the measurement error, which is driven (i) by the probability of an error in the birth location, (ii) by the difference in the exposure between the true and the reported place of birth and (iii) by the share of the variation in the exposure that is driven by temporal (vs spatial) variation. Environmental measures at a more granular, less aggregated level (e.g. at the precise coordinate level) are subject to a higher probability of error in the birth location, which will increase the measurement error and thus the attenuation bias. High levels of spatial correlation in the early life environment reduce the consequences of errors in individuals' birth location reports and with that the resulting attenuation bias. In contrast, low levels of spatial correlation increase attenuation bias, since

**Table 1 | Differences in siblings' birth location and their age gap**

| | Different birth location: | | | | | | | |
| --- | --- | --- | --- | --- | --- | --- | --- | --- |
| | (1) Parish | (2) District | (3) County | (4) d>0km | (5) d>5km | (6) d>10km | (7) d>30km | (8) d>50km |
| **Regression coefficients:** | | | | | | | | |
| Age gap (years) | 0.009 | 0.009 | 0.007 | 0.012 | 0.010 | 0.007 | 0.007 | 0.006 |
| | [0.007,0.012] | [0.007,0.011] | [0.005,0.009] | [0.010,0.014] | [0.008,0.012] | [0.005,0.009] | [0.005,0.008] | [0.005,0.008] |
| Constant | 0.307 | 0.292 | 0.123 | 0.486 | 0.255 | 0.160 | 0.086 | 0.066 |
| | [0.295,0.319] | [0.280,0.304] | [0.115,0.132] | [0.474,0.499] | [0.243,0.266] | [0.150,0.169] | [0.078,0.093] | [0.059,0.073] |
| **Derived probabilities:** | | | | | | | | |
| $\hat{q}$ (move probability) | 0.009 | 0.009 | 0.007 | 0.012 | 0.010 | 0.007 | 0.007 | 0.006 |
| | [0.007,0.012] | [0.007,0.011] | [0.005,0.009] | [0.010,0.014] | [0.008,0.012] | [0.005,0.009] | [0.005,0.008] | [0.005,0.008] |
| $\hat{p}$ (error probability) | 0.168 | 0.159 | 0.064 | 0.283 | 0.137 | 0.083 | 0.044 | 0.034 |
| | [0.161,0.175] | [0.152,0.165] | [0.059,0.069] | [0.275,0.292] | [0.130,0.143] | [0.078,0.089] | [0.040,0.048] | [0.030,0.037] |
| N (sibling pairs) | 18,479 | 18,479 | 18,479 | 18,479 | 18,479 | 18,479 | 18,479 | 18,479 |

The sample is restricted to the first sibling pair observed in each family. 95% confidence intervals based on heteroskedasticity robust standard errors are shown in brackets. Standard errors for $\hat{p}$ were computed using the delta method. In two-sided significance tests, all regression coefficients and derived probabilities are statistically different from zero with $p < 10^{-13}$.

small errors in birth locations can have large consequences for the measurement error in the exposure variable. Temporal variation in the early life environment is not affected by the measurement error of birth locations, thus the larger the share of variation in the exposure that is driven by temporal (as opposed to spatial) variation, the smaller the susceptibility to the attenuation bias.

Sibling fixed effects models may be particularly vulnerable to attenuation bias as they rely on differences in exposure between siblings. Such differences in exposure may be due to siblings' differences in their location or date of birth. In cases where the exposure mainly captures spatial (rather than temporal) variation, sibling differences in the exposure will be largely driven by measurement error. Based on our estimates in Table 1 and the average age gap between siblings of 4.5 years, we conclude that approximately 80–85% of birth location differences are driven by misreporting and only 15–20% are due to house moves. In other words, variation due to measurement error will dominate true variation in the measured exposure when time variation in the exposure is low. With higher levels of time variation, the share of true variation in the measure of exposure will increase, and the attenuation bias will reduce.

We quantify the size of the attenuation bias using Monte Carlo simulations for a variety of exposures with varying levels of spatial correlation and temporal variation and for linear regression (ordinary least squares; OLS) as well as sibling fixed effect specifications. Figure 2 shows the size of the attenuation bias for simulated spatial data at the district-of-birth level (we repeat this exercise at coordinate and parish levels of geographical aggregation in Supplementary Results 4). Panel a gives the percentage attenuation bias of the slope coefficient in a bivariate linear regression. For exposure variables without spatial correlation and time variation, the bias is predicted to be approximately 16.5%. As the spatial correlation increases, the consequences of measurement error in the birth location for the exposure variable are reduced and the attenuation bias shrinks. As the share of time variation increases, a smaller share of the overall variance in the exposure is subject to measurement error and the attenuation bias is reduced. Exposures with 100% variance due to time variation are no longer subject to any bias, as spatial measurement errors no longer have any consequences on the exposure. The maximum bias for exposures at the birth coordinate (Supplementary Fig. 4) and parish level (Supplementary Fig. 5) is 28% and 17%, respectively, and again the bias decreases as spatial correlation and time variation of the exposure increases.

In Fig. 3 we present the simulated attenuation bias for examples of previously studied or otherwise relevant district-level exposure variables with different levels of spatial and temporal variation (from alternative data sources; see note to figure), where each exposure has been standardised to have mean zero and standard deviation one. The predicted bias of OLS estimates ranges from 5.6% for the exposure to the infant mortality rate during the first year of life to 23% for the time-invariant share of social class III from the 1951 census. Note that the bias of OLS estimates for some of these examples exceeds the bias for simulated district-level exposures in Fig. 2. While the simulated exposures assume identical distributions of exposures for sibling pairs reporting the same birth location and sibling pairs who do not, this does not hold for all examples and can lead to a larger attenuation bias if the variance of an exposure is larger among sibling pairs reporting different birth locations.

Panel b of Fig. 2 reports the attenuation bias in sibling fixed effect models for simulated district-level exposures. We find a very large attenuation bias of 87–88% when the exposure variable does not vary over time. For data at the birth coordinate (Supplementary Fig. 4) and parish level (Supplementary Fig. 5) the bias is up to 90% and 88%, respectively. Even an increase in spatial correlation does not reduce this bias substantially in the absence of any time variation, since the true variation in the exposure from household moves and the false variation from measurement error decrease at the same rate. For

### a Ordinary least squares

**Variance share from time variation**

| Spatial autocorrelation ($\rho$) | 0% | 20% | 40% | 60% | 80% | 100% |
|---|---|---|---|---|---|---|
| 0.000 | 16.5 | 13.2 | 9.9 | 6.6 | 3.3 | 0.0 |
| 0.050 | 15.0 | 12.0 | 9.0 | 6.0 | 3.0 | 0.0 |
| 0.100 | 15.7 | 12.5 | 9.4 | 6.3 | 3.1 | 0.0 |
| 0.150 | 15.1 | 12.1 | 9.1 | 6.1 | 3.0 | 0.0 |
| 0.200 | 14.6 | 11.6 | 8.7 | 5.8 | 2.9 | 0.0 |
| 0.250 | 14.8 | 11.8 | 8.9 | 5.9 | 2.9 | 0.0 |
| 0.300 | 15.0 | 12.0 | 9.0 | 6.0 | 3.0 | 0.0 |
| 0.350 | 14.4 | 11.6 | 8.7 | 5.8 | 2.9 | 0.0 |
| 0.400 | 14.9 | 12.0 | 9.0 | 6.0 | 3.0 | 0.0 |
| 0.450 | 13.4 | 10.7 | 8.0 | 5.4 | 2.7 | 0.0 |
| 0.500 | 13.2 | 10.5 | 7.9 | 5.2 | 2.6 | 0.0 |
| 0.550 | 13.7 | 10.9 | 8.2 | 5.5 | 2.7 | 0.0 |
| 0.600 | 13.3 | 10.7 | 8.0 | 5.3 | 2.7 | 0.0 |
| 0.650 | 13.5 | 10.8 | 8.0 | 5.3 | 2.6 | 0.0 |
| 0.700 | 11.9 | 9.5 | 7.2 | 4.8 | 2.4 | 0.0 |
| 0.750 | 12.6 | 10.1 | 7.6 | 5.0 | 2.5 | 0.0 |
| 0.800 | 12.3 | 9.8 | 7.4 | 4.9 | 2.4 | 0.0 |
| 0.850 | 11.3 | 9.1 | 6.9 | 4.6 | 2.3 | 0.0 |
| 0.900 | 10.3 | 8.2 | 6.2 | 4.1 | 2.1 | 0.0 |
| 0.950 | 8.2 | 6.6 | 5.0 | 3.3 | 1.7 | 0.0 |
| 0.975 | 6.8 | 5.4 | 4.1 | 2.7 | 1.4 | 0.0 |

### b Sibling fixes effects

**Variance share from time variation**

| Spatial autocorrelation ($\rho$) | 0% | 20% | 40% | 60% | 80% | 100% |
|---|---|---|---|---|---|---|
| 0.000 | 87.7 | 52.0 | 31.0 | 17.2 | 7.3 | 0.0 |
| 0.050 | 87.8 | 49.8 | 29.0 | 15.8 | 6.6 | 0.0 |
| 0.100 | 87.6 | 50.5 | 29.6 | 16.2 | 6.8 | 0.0 |
| 0.150 | 87.7 | 50.1 | 29.2 | 15.9 | 6.7 | 0.0 |
| 0.200 | 87.8 | 49.2 | 28.5 | 15.5 | 6.5 | 0.0 |
| 0.250 | 87.7 | 49.4 | 28.7 | 15.6 | 6.6 | 0.0 |
| 0.300 | 87.7 | 50.0 | 29.1 | 15.9 | 6.7 | 0.0 |
| 0.350 | 87.7 | 49.1 | 28.5 | 15.5 | 6.5 | 0.0 |
| 0.400 | 87.9 | 49.9 | 29.2 | 16.0 | 6.8 | 0.0 |
| 0.450 | 87.7 | 47.7 | 27.1 | 14.5 | 6.0 | 0.0 |
| 0.500 | 87.7 | 46.7 | 26.3 | 14.1 | 5.9 | 0.0 |
| 0.550 | 87.7 | 48.1 | 27.5 | 14.8 | 6.2 | 0.0 |
| 0.600 | 87.7 | 47.4 | 26.9 | 14.4 | 6.0 | 0.0 |
| 0.650 | 87.9 | 47.3 | 26.8 | 14.4 | 6.0 | 0.0 |
| 0.700 | 87.6 | 45.3 | 25.1 | 13.3 | 5.5 | 0.0 |
| 0.750 | 87.6 | 46.3 | 25.9 | 13.8 | 5.7 | 0.0 |
| 0.800 | 87.6 | 45.2 | 25.1 | 13.3 | 5.5 | 0.0 |
| 0.850 | 87.4 | 43.8 | 24.0 | 12.7 | 5.2 | 0.0 |
| 0.900 | 87.4 | 42.2 | 22.6 | 11.7 | 4.8 | 0.0 |
| 0.950 | 87.3 | 37.0 | 18.9 | 9.6 | 3.9 | 0.0 |
| 0.975 | 87.4 | 32.5 | 15.9 | 7.8 | 3.1 | 0.0 |

**Fig. 2 | Attenuation bias for district-level data with different levels of spatial correlation and time variation. a** Bias in ordinary least squares estimations, **b** bias in sibling fixed effects estimations. The attenuation bias values shown are the mean bias (in %) from simulations of OLS and sibling fixed effects estimations with $r = 1000$ repetitions and sample sizes of $n = 36,940$ individuals each. For each level of spatial autocorrelation ($\rho$), ten district-level variables were simulated and merged into the sibling sample. The district-level spatial variables were combined with normally distributed year–month of birth fixed effects to simulate time-varying spatial exposures. The columns of the tables correspond to different ratios of spatial to temporal variation when simulating the exposure variable, as indicated by the share of the exposure variance due to time variation. Each simulated variable was then used in 100 simulations of the attenuation bias based on an error probability for the district of birth $p = 0.158$ and a move probability $q = 0.009$. Source data are provided as a source data file.

exposures that do vary over time and space, an increase in spatial correlation does reduce the bias. Furthermore, increases in the temporal variation reduce the bias substantially, shrinking it to zero for exposures driven solely by time variation. For the examples of district-level exposure variables (Fig. 3), the attenuation bias in a sibling fixed effects model is substantially larger than in bivariate linear regressions. The predicted bias ranges from 10% for exposure to highly time-varying measles rates to 87–88% for the census-based demographic measures as they are (by construction) time-invariant. These findings are in line with the measurement error literature[22,23] which show that fixed effects estimations may aggravate the attenuation bias, especially in cases where the signal is highly correlated over time (i.e. little time variation in the true exposure) but the error is not (i.e. no or little correlation between siblings' birth location errors).

### Bias in analyses controlling for district of birth fixed effects

Even if the variable of interest is measured without error, the corresponding regression coefficient may be subject to omitted variable bias when control variables (including birth location fixed effects) are subject to measurement error, thereby omitting part of the true control variable[24–26]. In other words, measurement error in the birth location data may lead to bias in the coefficient of interest in analyses that rely on birth location fixed effects as control variables[27]. This bias differs in important aspects from the attenuation bias when the variable of interest (i.e. not the control variable) is measured with error. In the case of classical measurement error in a control variable, the resulting "partially omitted variable bias" in the variable of interest will be smaller in magnitude and in the same direction as the bias from fully omitting the control variable. Thus, measurement error of control variables will not necessarily cause attenuation bias in the coefficient of interest but indeed can lead to bias away from zero. Unlike classical attenuation bias, the "partially omitted variable bias" is not proportional to the coefficient of interest and can therefore arise even when the variable of interest has no impact on the outcome.

We use Monte Carlo simulations to quantify the size of this bias in estimations controlling for district of birth fixed effects when the variable of interest is correctly observed but the district of birth is subject to measurement error. Supplementary Fig. 6 shows the size of the bias for different spatial autocorrelations ($\rho$) in the district fixed effects and different correlations between the fixed effects and the variable of interest. Our simulations show the bias to be proportional to the ratio of the standard deviation of the district fixed effects to that of the variable of interest ($\sigma_\mu/\sigma_X$). We therefore pool the simulation results for different standard deviations and express the bias in units of $\sigma_\mu/\sigma_X$.

The direction of the bias corresponds to the sign of the correlation between the variable of interest and the fixed effects: if they are positively (negatively) correlated, the bias is positive (negative). Similar to the bias from fully omitted control variables, the magnitude of the bias is increasing in the correlation of the variable of interest and the control variable, in this case, the district fixed effects. A higher spatial autocorrelation in the district fixed effects reduces the size of the bias, with errors in the district of birth having smaller consequences for the fixed effects.

Hence, in summary, our simulations show that measurement error in birth locations may lead to substantial bias in regressions that control for place of birth fixed effects, even when the variable of interest is measured accurately. The bias will be particularly large in the presence of substantial fixed effects relative to the variation in the variable of interest (i.e. a large value for $\sigma_\mu/\sigma_X$), when the variable of interest is strongly correlated with the fixed effects and when there is little spatial correlation in the fixed effects.

### Consequences of discordance in siblings' birth location for the spatial correlation of genetic principal components

Principal components of genotype data are spatially correlated within the United Kingdom[4,7]. Measurement error in birth location data and household mobility is therefore expected to affect the strength of

**Estimation**

| Exposure variables | OLS | Sibling FE |
|---|---|---|
| **Disease rates - 1st year of life:** | | |
| Diphtheria rate | 6.6 | 12.3 |
| Measles rate | 8.7 | 10.2 |
| Nonparalytic polio rate | 11.9 | 14.6 |
| Paralytic polio rate | 12.5 | 14.4 |
| Pneumonia rate | 11.2 | 33.2 |
| Respiratory tuberculosis rate | 7.9 | 20.6 |
| Scarlet fever rate | 6.6 | 11.8 |
| Whooping cough rate | 9.5 | 14.4 |
| **Demographics - 1st year of life:** | | |
| Birth rate | 11.3 | 30.8 |
| Death rate | 16.6 | 48.9 |
| Illegitimacy rate | 11.9 | 34.4 |
| Infant mortality rate | 5.6 | 16.1 |
| Stillbirth rate | 15.0 | 20.5 |
| **Demographics - 1951 census:** | | |
| Share housing density 0 - 1 | 8.1 | 87.5 |
| Share housing density 1 - 1.5 | 10.6 | 87.7 |
| Share housing density 1.5 - 2 | 7.5 | 87.6 |
| Share housing density 2 - 3 | 7.8 | 87.5 |
| Share housing density 3+ | 10.0 | 88.0 |
| Share in social class I | 15.0 | 88.0 |
| Share in social class II | 16.6 | 87.5 |
| Share in social class III | 23.4 | 88.2 |
| Share in social class IV | 15.5 | 87.9 |
| Share in social class V | 11.9 | 87.6 |
| Share left FT education at 0-14 | 11.6 | 87.7 |
| Share left FT education at 15 | 11.4 | 87.5 |
| Share left FT education at 16 | 11.3 | 87.6 |
| Share left FT education at 17-19 | 12.3 | 87.3 |
| Share left FT education at 20+ | 18.3 | 87.8 |

**Fig. 3 | Attenuation bias for different district-level exposure variables.** The attenuation bias values shown are the mean bias (in %) from simulations of OLS and sibling fixed effects estimations with $r = 1000$ repetitions based on an error probability for the district of birth $p = 0.158$ and a move probability $q = 0.009$. The individual sample sizes used in the simulations differ between the different exposure variables and are provided in the source data. Disease data are from the Registrar General's Weekly Reports[39,40], demographic data are from the Registrar General's Statistical Review of England and Wales[41] and 1951 census[34,42]. Source data are provided as a source data file.

these spatial correlations. We explore this in the following. Supplementary Fig. 7 confirms the strong levels of spatial correlation for the first five principal components in the UKB (based on principal component analysis for a homogeneous population of white British UKB respondents; Moran's $I > 0.8$ at the district level). We find much less spatial correlation for the sixth principal component and therefore do not focus our discussion on the latter.

The vertical axis of Fig. 4 shows the correlation of each genetic principal component between individuals in the sibling sample and the mean among individuals in the non-sibling sample who reported the same birth location, including their 95% error bars. We estimate these correlations separately for siblings with different levels of discordance in their reported birth location, measured along the horizontal axis. Correlations between siblings without any discordance in birth location and others reporting the same birth location are above 0.35 for the first five principal components, with some as high as 0.6. As the distance between siblings' reported birth location increases in Fig. 4, the correlation with others reporting the same birth location reduces.

Indeed, when comparing siblings who reported being born more than 200 km apart (3.6% of sibling pairs) to those without any discordance, the correlation coefficients decrease by more than 40% for all five spatially structured principal components. These are significant differences for each of the first five principal components. Note that the figures look similar when we separately plot them for first- and later-borns. These results illustrate the impact of household mobility and measurement error on the estimation of the spatial structure of genetic data.

**Possible solutions**

What can we do to investigate the robustness of estimates that exploit the retrospectively reported birth location data? Although any solution will depend on the empirical analysis and the sample of interest, we here discuss three possible solutions, highlighting their advantages as well as potential drawbacks. First, for analyses that do not focus on within-sibling variation, a potential sensitivity analysis is to limit the sample to siblings who reported the same birth location. While this

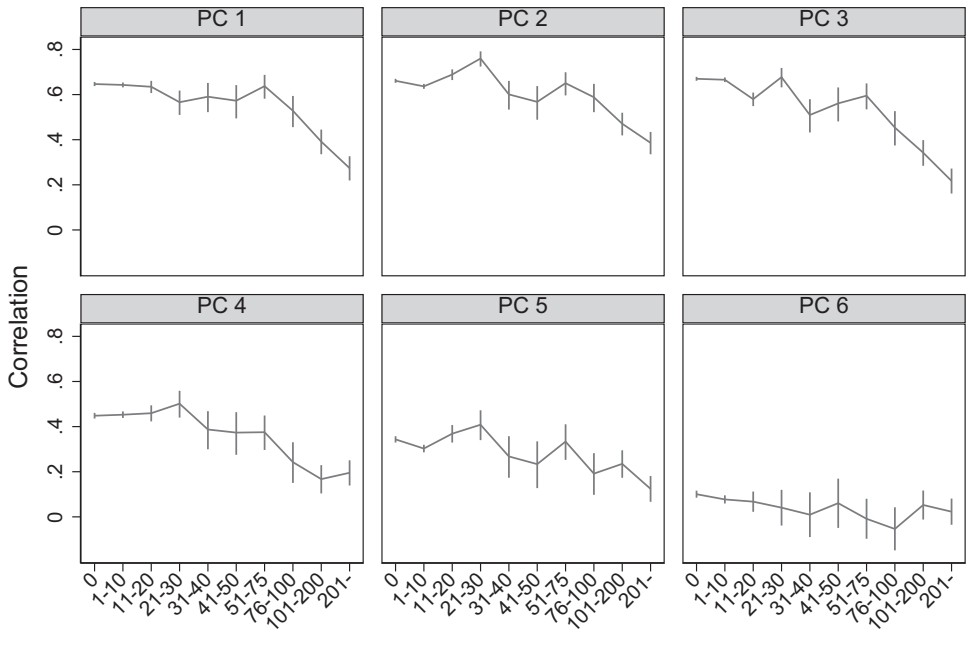

Distance to sibling's birth location (km)

**Fig. 4 | Correlation of siblings' genetic principal components with others reporting the same birth location.** The figures show the correlation between the genetic principal components of individuals in our sibling sample (n = 33,229 individuals) and the mean genetic principal components of non-sibling UKB participants who reported the same birth coordinates. Vertical bars represent 95% confidence intervals. Correlations are shown separately for the first six genetic principal components (PC1–PC6) and for distances between the reported birth locations of the individuals and their siblings. Principal components are based on principal component analysis conducted on unrelated white british individuals in the UKB. SNPs were filtered based on minor allele frequency >0.01 and clumped for linkage disequilibrium based on minor allele frequency ($R^2 > 0.1$). Long-range linkage disequilibrium regions were removed. Source data are provided as a source data file.

means omitting non-siblings as well as siblings who experienced house moves in childhood from the analysis, it allows focusing on a subsample with more reliable (i.e. with less error in the) birth location data. In Supplementary Analysis 1 we apply this sensitivity check to replicate some of the analysis of a high-profile study that uses birth location data in the UKB[4].

Second, one could use siblings' reported birth location as an instrument for the imperfectly measured birth location of an individual. This can be used to address the measurement error and resulting attenuation bias in the estimates, e.g. using an Obviously Related Instrumental Variables (ORIV[28]), approach similar to research using this to deal with measurement error in polygenic indices[29]. The downside, however, is that the measurement error in retrospectively recorded birth locations is non-classical. In such cases, ORIV estimates are unlikely to overcome the bias[23]. Indeed, our simulations show that ORIV estimates remain biased.

Finally, if the interest lies in estimating causal gene–environment ($G \times E$) interactions (an increasingly popular application of sibling data since the random within-sibling variation in genes allows a causal interpretation[30]) we recommend using an alternative approach to a sibling fixed effects specification due to the large attenuation bias in these models. More specifically, if the environmental measure (based on birth date and place) is exogenous both within and between siblings (i.e. when the sibling fixed effects are not required for a causal interpretation of the environmental effect), it is better to use the deviation of a sibling's genetic measure from the mean of the sibling pair (or group) as an exogenous source of genetic variation in the sibling sample[14,21], but use both within- and between-sibling variation in the environmental measure to avoid attenuation bias from measurement error in the birth location. Instead of using this sibling mean deviation, one can alternatively control for the (imputed) parental genotypes[31].

## Discussion

A large number of data surveys ask participants to retrospectively record their residential location of birth. This information has been used in many research papers across a wide range of empirical applications. Despite their frequent use, there is little information on how accurate these data are, especially since they often rely on individuals' accurate 30+ year recall. To address this, we explore the accuracy of retrospectively recorded birth location data by studying the sample of full siblings in the UKB. Our paper makes three distinct contributions. First, assuming that siblings grew up together, our analysis allows us to decompose the discordance in birth location into two components and quantify the importance of both: household moves and measurement error. Our estimates suggest that household mobility during the early childhood period was low, with an estimated average annual move probability of 1.2%. Although our results show substantial measurement error in participants' birth location, in the majority of cases, participants report nearby locations, meaning that most errors are of a short distance.

Our second contribution is to discuss the implications of measurement error in reported birth locations. We show that the consequences depend on what the birth location data are used for. We focus on the attenuation bias that results from the use of external data merged based on individuals' location and date of birth as explanatory variables in a regression. We quantify the size of this bias using Monte Carlo simulations for a variety of exposure variables with varying levels of spatial correlation and time variation, including disease exposures and demographic variables.

Since the majority of errors are of a short distance, analyses at the coordinate level are more strongly affected than analyses at the district or parish level. We show that, as the share of temporal variation as well as the spatial correlation in the variable of interest increases, attenuation bias reduces. Hence, the estimated impacts of exposures

that show substantial temporal variation, such as disease or infant mortality rates, are substantially less attenuated compared to the impact of exposures that are time-invariant, such as census-based measures of socio-economic composition. We show that sibling fixed effect specifications, only exploiting variation in the outcomes and variables of interest within families (between siblings), are particularly vulnerable to the attenuation bias, with the bias exceeding 80% when using time-invariant exposure variables. We thus recommend caution when using the UKB birth location data, especially when estimating sibling fixed effects models where the variable of interest has limited time variation and thus a low ratio of signal to error.

Our third contribution is to highlight possible solutions to the attenuation bias. Although these will depend on the sample of interest (e.g. siblings, non-siblings), the variation used (e.g. within vs between family variation) and the estimate of interest (e.g. the effect of environmental exposure at birth or gene–environment interplay), we highlight three possible sensitivity analyses. This will allow the researcher to explore the importance of the attenuation bias and robustness to alternative specifications.

More generally, the results in this paper highlight the impact of measurement error in birth location data on regression estimates that exploit this information. We show substantial attenuation bias that is a function of the spatial correlation and temporal variation of the variable of interest. Finally, we show that the measurement error also impacts on the spatial structure of the genetic data.

## Methods

### UK Biobank sibling sample
The UKB is a large-scale, mostly biomedical database of over 500,000 individuals living in the UK. It has received ethical approval from the NHS North West Centre for Research Ethics Committees (references: 11/NW/0382, 16/NW/0274, 21/NW/0157). We focus on the sample of full siblings, identified using their genetic data, which includes 41,441 UKB participants. The UKB did not explicitly sample families or households, but nevertheless does contain a substantial number of related individuals. Family relationships to other participants were not recorded in the interviews, but we can identify biological relatives based on their genetic relatedness. By using the kinship matrix provided by the UKB[32], we can identify related participants and derive their relationship. The kinship matrix contains relatives up to the third degree and was constructed using the KING software[33]. A kinship coefficient of approximately 0.25 (interval: 0.1770–0.3540) suggests that the individuals are either parent-offspring pairs or full siblings. One can then distinguish between these two types of relationships by using the identity by state ($IBS_0$) coefficient: a coefficient above 0.0012 suggests that the pair of individuals are siblings rather than parent and offspring.

The main variables of interest in our analysis are individuals' east and north coordinates of birth (field IDs 129 and 130), reported at a 1-km resolution. We restrict the sample to families with at least two siblings born in England, Wales and Scotland (3388 participants dropped) and use the birth coordinates to identify their parish, local government district (henceforth: district) and county of birth, using the regional boundaries in 1951[34,35]. Keeping the boundaries fixed over time ensures that any regional boundary changes cannot drive any differences in (e.g.) the parish or the district of birth. Our sibling sample covers 2518 parishes, 1363 districts and 98 counties. We focus on birth location data in the form of 1 km grid coordinates as well as historical parishes, districts and counties since most applications use information aggregated to these geographical identifiers (e.g. to merge external data or as control variables). Furthermore, analysing the actual grid coordinates allows us to examine errors before the mapping to historical areas may introduce additional processing errors.

Additionally, we restrict the sample to the oldest two siblings observed in each family (1095 participants dropped). Our final sample

comprises 36,958 siblings from 18,479 families. Supplementary Table 11 presents some descriptive statistics on our sibling sample, showing that the age gap between siblings varies between 0 and 27 years, with a mean of 4.5 years. The data comprises 227 pairs of twins, and 57.7% of the sibling sample are female. On average, siblings report being born 22 km apart, though this is highly skewed, ranging from 0 to 1060 km. The sibling sample is relatively similar in individual and district characteristics compared to the full UKB sample (see Supplementary Table 12), with 63–66% having an upper secondary qualification and 84–86% being born in an urban or municipal district.

### Polygenic indices
We explore the heterogeneity of our results with respect to the siblings' genetic predisposition for educational attainment. For this purpose, we use the PGI for education provided in the PGI repository[36] to split the sample into quartiles and below-/above-median sub-samples. Specifically, we use the single-trait PGI provided in the repository for (what they call) the first partition of the UKB which includes all siblings. This PGI is based on a discovery sample of $N = 984,323$ which includes the other two partitions of the UKB as well as other datasets such as 23andMe, AddHealth and HRS. In line with the current genetics literature, we restrict our sample in estimations involving PGIs to those of European ancestry. Thus the sample size reduces from 18,479 sibling pairs in our main estimations to 18,048 sibling pairs in estimations with the PGI. In this sample, the PGI for educational attainment has an incremental $R^2$ of 10.9%.

### Comparison of siblings' birth locations
We start by constructing a binary variable indicating whether the siblings reported being born in a different location. The variable $y_s$ equals one if either the north or east coordinates differ between the two full siblings of family $s$. We then estimate the following linear probability model:

$$P(y_s = 1 | agegap_s) = \alpha + \beta \times agegap_s \qquad (1)$$

where $agegap_s$ is the age gap (in years) between the two siblings in family $s$. Hence, this assumes that the discordance in reported birth location is driven by two processes: (1) 'true' differences due to house moves and (2) reporting/measurement error. The extent to which $agegap_s$ can explain variation in $y_s$ captures the former; the intercept $\alpha$ captures the latter. We explore potential non-linearities in this relationship in Supplementary Results 1. We furthermore estimate linear probability models as described by Equation (1) for a variety of alternative definitions of $y_s$: different parishes of birth, different districts of birth, different counties of birth and different birth locations that are more than 5/10/20/30/50 km apart.

Birth date variables in the UKB are recorded at the year–month level rather than at the daily level. This rounding introduces some classical measurement error to the age gap variable used in our estimations. The expected attenuation bias from this measurement error can be derived to be 0.01% and thus should not have any substantial impact on our estimates. Further measurement error in the reported birth date variable (and thus in the age gap between siblings) is likely to be small. Suggestive evidence of the reliability of the birth date and age gap variable is that we do not observe any sibling pairs with an age gap of 1–8 months, as would be expected based on the length of human gestation.

### Deriving probabilities of household moves and measurement error
We use the regression estimates of Equation (1) to derive the probability of a household move during childhood as well as the probability of measurement error in the location of birth. To derive these probabilities, we make the following assumptions:

**Assumption 1.** If both siblings in a sibling pair report the same birth location, we assume this is the true birth location for both.

Assumption 1 recognises that we cannot identify misreporting of birth locations if both siblings report the same incorrect birth locations. Ignoring these unlikely cases will downward bias the derived error probability. We furthermore cannot identify household moves in cases where the siblings incorrectly report the same birth location.

**Assumption 2.** We assume that biological siblings grew up in the same household.

Assumption 2 ignores adoptions shortly after birth and other events that may result in biological siblings growing up in different households. However, the frequency of such events in the UK during 1940–70 was likely low, and thus any resulting upward bias in the derived error probability will be small.

**Assumption 3.** We assume that the probability of a move between the birth of two siblings is a linear function of the age gap between them.

We empirically test Assumption 3 in Supplementary Results 1 by allowing for non-linearities in the relationship between the sibling age gap and the probability of house moves. While we find minor non-linearities, these only affect the derived error probabilities to a small degree.

**Assumption 4.** We assume that measurement error occurs randomly with the same probability for any participant and that these errors are independent within sibling pairs and independent of the age gap between siblings.

Assumption 4 is unlikely to hold in reality. Indeed, we show in Supplementary Results 2 that certain participant characteristics affect the probability of house moves and measurement errors. It is furthermore likely that error occurrence is positively correlated among siblings, but our data does not allow us to quantify this. Ignoring such a positive within-sibling correlation will downward bias the derived error probability. However, simulations show that a very strong sibling correlation is required to create substantial bias.

Under the above assumptions, the probability of a difference in the reported birth location between two siblings can be written as:

$$P(location_{sib1} \neq location_{sib2}) = P(move_{sib1,sib2}) + P(error_{sib1} \cup error_{sib2})$$
$$= q \times agegap_{sib1,sib2} + P(error_{sib1} \cup error_{sib2}) \quad (2)$$

where $q$ denotes the annual probability of a household moving to a different location. The probability of an error in the birth location of either sibling can be written as:

$$P(error_{sib1} \cup error_{sib2}) = P(error_{sib1}) + P(error_{sib2}) - P(error_{sib1} \cap error_{sib2})$$
$$= p + p - p^2 = 2p - p^2 \quad (3)$$

We use this to derive the probability $p$ of an error in the birth location of any respondent, defined as:

$$p = 1 - \sqrt{1 - P(error_{sib1} \cup error_{sib2})} \quad (4)$$

**Attenuation bias**

Measurement error in the birth location data may have consequences for the use of birth location-based explanatory variables in regression analyses. "Classical" measurement error in explanatory variables

causes coefficient estimates in a linear regression to be biased towards zero[23,37]. In Supplementary Note 2 we discuss this attenuation bias in the presence of classical measurement error. Our application is unlikely to show classical measurement error. For one, with the majority of siblings reporting the same birth location, there is a spike at zero. Even when ignoring the zeros, the normality of the measurement errors may not hold in our setting. In Supplementary Fig. 10, we show that the distribution of errors in our district-level simulations is approximately normal when spatial autocorrelations are small. Large levels of spatial correlation, however, can lead to a leptokurtic distribution. Furthermore, one would expect a negative correlation between the true explanatory variable and the measurement error: if an exposure is high (low) for the true birth location, it is more likely that the exposure in the misreported birth location is below (above) the true exposure level due to regression to the mean. We confirm this expectation in Supplementary Tables 13 and 14, which show large negative correlations between the true explanatory variable and the measurement error in our district-level simulations; in particular when focusing on observations with non-zero measurement error.

We use Monte Carlo simulations to quantify the size of the attenuation bias in linear regression (OLS) and sibling fixed effect estimations for simulated data at the coordinate, parish and district level with varying levels of spatial correlation and temporal variation, as well as for several district-level measures of disease exposure and demographics. The simulations are based on the birth location differences observed in the data and therefore do not assume classical measurement error.

**Simulation of time-varying spatially correlated data.** We begin by simulating time-invariant spatially correlated data at the district, parish and coordinate level based on a spatial autoregressive model with spatial autocorrelation parameter $\rho$ that ranges from −1 to 1 (if the spatial weighting matrix is row-standardised). A positive $\rho$ corresponds to spatial clustering, with larger values of $\rho$ indicating stronger spatial clustering. A negative $\rho$ indicates spatial dispersion and $\rho$ equals zero when there is no spatial autocorrelation. We use the `sim_sar` command of the `geostan` package (for parish- and district-level data) and the `powerWeights` command of the `spatialreg` package (for coordinate-level data) in R to simulate 10 variables $S_{a,\rho}$ for each spatial aggregation level $a$ (coordinate-, parish- and district-level) and spatial autocorrelation parameter $\rho \in [0.00, 0.05, 0.10, ..., 0.90, 0.95, 0.975]$. We simulate ten variables (rather than one) to ensure that our simulation results are not driven by an "outlier" in the spatial simulations, which is particularly important when the number of spatial units is relatively small.

To add time variation to the spatial data, we draw year–month of birth fixed effects from a standard normal distribution (without any temporal autocorrelation). We simulate a year–month of birth fixed effect variable $T$ for each of the simulated spatially correlated time-invariant variables $S_{a,\rho}$. Finally, we create time-varying spatially correlated variables for different shares of variance due to temporal variation $k \in [0.0, 0.2, 0.4, 0.6, 0.8, 1.0]$ by combining the standardised time-invariant spatial variables and the standardised birth date fixed effects as follows:

$$V_{a,\rho,k} = \sqrt{k}*T + \sqrt{1-k}*S_{a,\rho} \quad (5)$$

We standardise the resulting time-varying spatially correlated $V_{a,\rho,k}$ to have a mean of zero and a standard deviation of one.

**Simulation of attenuation bias.** In our simulations of the attenuation bias, the simulated time-varying spatially correlated variables $V_{a,\rho}$ as well as a variety of standardised district-level measures of disease exposure and demographics are merged with the sibling sample used

in our main estimations to construct for each simulation run (i) the observed exposures $X^*_{s,i}$ and (ii) the true exposures $X_{s,i}$.

We begin by merging the time-varying area-level variables to the sibling sample based on each individual's reported birth date and birth location; the resulting variable is defined as $X_{t,ownloc}$, where ownloc indicates the individual's reported birth location. Additionally, we merge these variables based on each individual's reported birth date and their sibling's reported birth location ($X_{t,sibloc}$), as well as the geographic midpoint between the two birth locations reported by the two siblings ($X_{t,midloc}$). If the geographic midpoint between the two birth locations is not on land, the closest UK land location to the midpoint is used.

For each sibling pair with different birth locations (defined at the coordinate, parish or district level) we compute the predicted probability of an error in the reported birth location conditional on different locations being reported:

$$\hat{P}(\text{error}_{sib1} \cup \text{error}_{sib2}|\text{location}_{sib1} \neq \text{location}_{sib2}) = \frac{\hat{P}(\text{error}_{sib1} \cup \text{error}_{sib2})}{\hat{P}(\text{location}_{sib1} \neq \text{location}_{sib2})}$$
$$= \frac{2\hat{p} - \hat{p}^2}{\hat{q}*\text{agegap}_{sib1,sib2} + 2\hat{p} - \hat{p}^2} \quad (6)$$

based on the estimates $\hat{p}$ and $\hat{q}$ from the main analysis (Table 1) and the siblings' age gap.

For each variable we then simulate the attenuation bias using 1000 repetitions. For the simulated time-varying spatially correlated variables $V_{a,\rho,k}$, we run 100 repetitions for each variable. With ten variables for each level of aggregation $a$, autocorrelation $\rho$ and temporal variance share $k$, this corresponds to 1000 simulations. Each repetition of the Monte Carlo simulation proceeds as follows: For sibling pairs with different reported birth coordinates/districts/parishes, we draw a Bernoulli random variable $E_s$ indicating whether an error occurred ($E_s = 1$) using the error probability $\hat{P}(\text{error}_{sib1} \cup \text{error}_{sib2}|\text{location}_{sib1} \neq \text{location}_{sib2})$ computed above. For sibling pairs with the same reported birth coordinates/district/parish, we set $E_s = 0$. A second Bernoulli variable $B_s$ is then drawn for those sibling pairs with $E_s = 1$ to indicate whether both sibling birth locations are subject to an error, based on the conditional probability $\hat{P}(\text{error}_{sib1} \cap \text{error}_{sib2}|\text{error}_{sib1} \cup \text{error}_{sib2}) = \hat{p}^2/(2\hat{p} - \hat{p}^2)$. If only one of the birth locations is incorrect ($E_s = 1$, $B_s = 0$), then one of the two siblings is chosen at random for the error.

The true exposure in our simulations is defined as follows:

$$X_{s,i} = \begin{cases} X_{t,sibloc} & \text{if error in this sibling only} \\ X_{t,midloc} & \text{if error in both siblings} \\ X_{t,ownloc} & \text{otherwise} \end{cases} \quad (7)$$

Irrespective of any errors to the reported birth location, the exposure $X_{s,i}$ is always defined based on the individual's reported date of birth $t$. If only one sibling in a sibling pair is simulated to have an incorrect birth location, their sibling's reported birth location *sibloc* is used to compute the exposure. If both siblings in a sibling pair are simulated to have an incorrect birth location, we use the mid-point *midloc* between their reported birth locations. In the absence of any information on the true birth location in these cases, the midpoint between the two reported locations sets a lower bound on the average geographic distance between the true and the observed birth locations. For all individuals who are not simulated to have an incorrect birth location, we use their own reported birth location *ownloc*.

The observed exposure in our simulations is defined as

$$X^*_{s,i} = X_{t,ownloc} \quad (8)$$

for all individuals and thus subject to measurement error due to misreported birth locations.

We simulate an outcome $Y_{s,i}$ which is affected by the true exposure as follows:

$$Y_{s,i} = X_{s,i} + \varepsilon_{s,i} \quad \text{with} \quad \varepsilon_{s,i} \sim N(0,1) \quad (9)$$

The exposures $X_{s,i}$ in our simulations are standardised to have a mean of zero and a standard deviation of one. We then calculate the OLS attenuation bias by comparing the coefficients estimated for the following two OLS equations:

$$Y_{s,i} = \alpha_1 + \beta_1 X_{s,i} + e_{1,s,i} \quad (10)$$

$$Y_{s,i} = \alpha_2 + \beta_2 X^*_{s,i} + e_{2,s,i} \quad (11)$$

The difference between $\hat{\beta_1}$ in the estimations using the true exposure and $\hat{\beta_2}$ in the estimations using the observed exposure is the OLS attenuation bias. Similarly, we calculate the attenuation bias in sibling fixed effects estimations by comparing the coefficient estimates $\hat{\gamma_1}$ and $\hat{\gamma_2}$ for the following two equations:

$$Y_{s,i} = \mu_{s,1} + \gamma_1 X_{s,i} + u_{1,s,i} \quad (12)$$

$$Y_{s,i} = \mu_{s,2} + \gamma_2 X^*_{s,i} + u_{2,s,i} \quad (13)$$

## Bias in analyses controlling for district of birth fixed effects

We use Monte Carlo simulations to quantify the size of the bias in estimations controlling for district of birth fixed effects when the variable of interest is correctly observed but the district of birth is subject to measurement error. We simulate data at the district level with varying levels of spatial correlation and correlations between the variable of interest and the district of birth fixed effects. In our simulations, we consider the following model:

$$Y_{i,d} = \beta X_{i,d} + \mu_d + \varepsilon_{i,d} \quad \text{with} \quad \varepsilon_{i,d} \sim N(0,1) \quad (14)$$

where outcome $Y_{i,d}$ is a function of the variable of interest $X_{i,d}$ and the district of birth fixed effects $\mu_d$. We furthermore model the correlation between $X_{i,d}$ and $\mu_d$ as follows:

$$X_{i,d} = (r\frac{\mu_d}{\sigma_\mu} + \sqrt{1 - r^2}\eta_i)*\sigma_x \quad \text{with} \quad \eta_i \sim N(0,1) \quad (15)$$

where $r$ is the correlation between $X_{i,d}$ and $\mu_d$, $\sigma_\mu$ is the standard deviation of the fixed effects $\mu_d$ and $\sigma_x$ is the standard deviation of $X_{i,d}$. In our simulations we examine the bias for different correlations $r \in [-0.95, -0.75, -0.50, -0.25, 0.00, 0.25, 0.50, 0.75, 0.95]$ and ratios of the standard deviations $\sigma_\mu/\sigma_x \in [0.1, 0.5, 1.0, 5.0]$. If there is measurement error in individuals' district of birth $d$, estimates of $\mu_d$ will be attenuated. Thus not all district-level variation in $Y_{i,d}$ will be controlled for and estimates of $\beta$ will be subject to omitted variable bias if $r \neq 0$.

We simulate spatially correlated district of birth fixed effects $\mu_{d,\rho}$ based on a spatial autoregressive model with spatial autocorrelation parameter $\rho \in [0.00, 0.05, 0.10, ..., 0.90, 0.95, 0.975]$. These simulated districts of birth fixed effects are then merged with the sibling sample used in our main estimations based on each individual's reported birth location ($\mu_{ownloc}$), the birth location reported by each individual's sibling ($\mu_{sibloc}$), as well as the geographic midpoint between the two birth locations ($\mu_{midloc}$).

For each sibling pair with different districts of birth, we compute the predicted probability of an error conditional on different districts

being reported:

$$\hat{P}(\text{error}_{sib1} \cup \text{error}_{sib2}|\text{district}_{sib1} \neq \text{district}_{sib2}) = \frac{\hat{P}(\text{error}_{sib1} \cup \text{error}_{sib2})}{\hat{P}(\text{district}_{sib1} \neq \text{district}_{sib2})}$$

$$= \frac{2\hat{p} - \hat{p}^2}{\hat{q}^*\text{agegap}_{sib1,sib2} + 2\hat{p} - \hat{p}^2}$$

(16)

based on the estimates $\hat{p}$ and $\hat{q}$ from the main analysis (Table 1) and the siblings' age gap.

Each repetition of the Monte Carlo simulation proceeds as follows: For sibling pairs with different reported birth districts, we draw a Bernoulli random variable $E_s$ indicating whether an error occurred ($E_s = 1$) using the error probability $\hat{P}(\text{error}_{sib1} \cup \text{error}_{sib2}|\text{district}_{sib1} \neq \text{district}_{sib2})$ computed above. For sibling pairs with the same reported birth district, we set $E_s = 0$. A second Bernoulli variable $B_s$ is then drawn for those sibling pairs with $E_s = 1$ to indicate whether both sibling birth districts are subject to an error, based on the conditional probability $\hat{P}(\text{error}_{sib1} \cap \text{error}_{sib2}|\text{error}_{sib1} \cup \text{error}_{sib2}) = \hat{p}^2/(2\hat{p} - \hat{p}^2)$. If only one of the birth districts is incorrect ($E_s = 1, B_s = 0$), then one of the two siblings is chosen at random for the error.

The true district of birth fixed effects in our simulations are defined as follows:

$$\mu_d = \begin{cases} \mu_{sibloc} & \text{if error in this sibling only} \\ \mu_{midloc} & \text{if error in both siblings} \\ \mu_{ownloc} & \text{otherwise} \end{cases}$$

(17)

If only one sibling in a sibling pair is simulated to have an incorrect district of birth, their sibling's reported birth location *sibloc* is used to compute their fixed effect. If both siblings in a sibling pair are simulated to have an incorrect district of birth, we use the mid-point *midloc* between their reported birth locations. In the absence of any information on the true birth location in these cases, the midpoint between the two reported locations sets a lower bound on the average geographic distance between the true and the observed birth locations. For all individuals who are not simulated to have an incorrect district of birth, we use their own reported birth location *ownloc*.

The observed district of birth in our simulations is defined as

$$d_i^* = d_{ownloc}$$

(18)

for all individuals and thus is subject to measurement error due to misreported birth locations.

We then calculate the bias by comparing the coefficients estimated for the following two fixed effects equations:

$$Y_{i,d} = \beta_1 X_{i,d} + \gamma_d + e_{1,i,d}$$

(19)

$$Y_{i,d} = \beta_2 X_{i,d} + \gamma_{d^*} + e_{2,i,d}$$

(20)

The difference between $\widehat{\beta_2}$ in the estimations using the observed districts of birth and $\widehat{\beta_1}$ in the estimations using the true districts of birth is the bias. We simulate the bias using 250 repetitions for each $\rho$, $r$ and $\sigma_\mu/\sigma_x$ (25 repetitions for each of the ten simulated district fixed effects variables $\mu_{d,\rho}$).

### Principal component analysis

We examine the impact of measurement error in the reported birth locations and of household mobility on the strength of spatial correlations in genetic principal components. For these analyses, we conduct principal component analyses using the `big_randomSVD` command of the `bigstatsr` package. We restrict the sample to unrelated white British individuals in the UKB and remove genetic outliers[38], resulting in a sample size of $N = 276{,}279$. SNPs are filtered based on minor allele frequency >0.01 and clumped for linkage disequilibrium based on minor allele frequency ($R^2 > 0.1$). Long-range linkage disequilibrium regions are removed. This results in a set of 108,251 SNPs used in the principal component analysis. We predict the resulting principal component vectors in the estimation sample of unrelated white British individuals described above, as well as in the sibling sample ($N = 33{,}741$ after removing any genetic outliers).

### Analysis of repeat birth location reports

A subset of UKB participants have reported their birth locations more than once. 18,975 participants gave an additional birth location report during a first repeat assessment visit in 2012/13 and 9374 participants during imaging visits between 2014 and 2022. In a supplementary analysis, we examine the share of participants who report different birth locations in these follow-up interviews compared to their initial interviews. For participants with more than one repeat birth location report, we only compare the first repeat report with the initial interview. Our sample for this comparison comprises 24,838 participants. While this comparison does not allow us to derive probabilities of household moves, we can derive the probability of measurement error in the location of birth in a similar way to the sibling comparisons described in the 'Methods' above. To derive the error probability we make the following assumptions:

**Assumption 1'.** If a participant reports the same birth location in the initial and follow-up interview, we assume this is their true birth location.

**Assumption 2'.** We assume that measurement error occurs randomly with the same probability for any participant and interview, and that these errors occur independently in the two interviews of the same participant.

Under these assumptions, the probability of a difference in the reported birth location between a participant's two interviews can be written as:

$$P(\text{location}_{i,t=1} \neq \text{location}_{i,t=2}) = P(\text{error}_{i,t=1} \cup \text{error}_{i,t=2})$$
$$= P(\text{error}_{i,t=1}) + P(\text{error}_{i,t=2}) - P(\text{error}_{i,t=1} \cap \text{error}_{i,t=2})$$
$$= p + p - p^2 = 2p - p^2$$

(21)

We use this to derive the probability $p$ of an error in the birth location of any respondent, defined as:

$$p = 1 - \sqrt{1 - P(\text{error}_{i,t=1} \cup \text{error}_{i,t=2})}$$

(22)

### Reporting summary

Further information on research design is available in the Nature Portfolio Reporting Summary linked to this article.

## Data availability

This research has been conducted using data from UK Biobank[32], a major biomedical database (Project ID: 74002). UK Biobank data are available following an application procedure described at https://www.ukbiobank.ac.uk/enable-your-research. This research is furthermore based on data provided through www.VisionofBritain.org.uk[34] and uses historical material which is copyright of the Great Britain Historical GIS Project and the University of Portsmouth. Data on boundaries of historic parishes, districts and counties in 1951 have previously been made available by the Vision of Britain project. For details on the

current or future availability of the boundary data, please see https://www.visionofbritain.org.uk/data/. District-level demographic data from the Registrar General's Statistical Review of England and Wales[41] are available via the UK data service at https://doi.org/10.5255/UKDA-SN-9035-1. District-level data on housing density, social class and education from the 1951 census[42] are available via the UK data service at https://doi.org/10.5255/UKDA-SN-4554-2, https://doi.org/10.5255/UKDA-SN-4561-2 and https://doi.org/10.5255/UKDA-SN-4552-2. Disease data are from the Registrar General's Weekly Reports[39,40]. Source data for figures are provided with this paper.

## Code availability
The code to conduct the data analyses is available under https://doi.org/10.5281/zenodo.10631529[43].

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

## Acknowledgements

This research has been conducted using the UK Biobank Resource under Application Number 74002. We gratefully acknowledge funding of this project from the European Research Council (ERC) under the European Union's Horizon 2020 research and innovation program (grant agreement no. 851725; S.v.H. and N.V.). We thank the GEIGHEI and ESSGN project members and participants at the European Social Science Genomics Network (ESSGN) Conference for their helpful comments.

## Author contributions

S.v.H. and N.V. conceived and designed the study. N.V. performed the empirical analysis and simulations. S.v.H. and N.V. contributed to writing the manuscript and supplementary information.

## Competing interests

The authors declare no competing interests.
