## [Peer Review File · Nature Communications]

An analysis of the accuracy of retrospective birth location recall using sibling dataREVIEWER COMMENTS

Reviewer #1 (Remarks to the Author):

Review of: "Accuracy of retrospective birth location data – An analysis based on siblings"

Prof. Von Hinke & Dr. Vitt present what can only be described as an essential public evaluation of the quality of birthplace reports in UK Biobank, and to an extent of retroactively reported birthplace in general. I have a series of what I feel are important comments and remarks but in general the value of the work is obvious and the quality high.

I have a lot of comments because I kind of love this article and see its utility to the field, I urge the editor not to mistake the quantity of feedback for a negative overall assessment.

Issue 1, taxonomy:

The authors study the frequently with which birthplaces are misreported, in their intro they go on to state that there are two causes why siblings could have different birthplaces:

"Assuming that the siblings grew up together, their location of birth can differ for two reasons. First, the family may have moved house between the births of their two or more children (i.e., a 'true' change in their birth location). Second, the location of birth may have been misreported by one or more children (i.e., an 'error' in their birth location)."

The authors then say that in fact the second reason is a composite of 2 reasons:

"The latter can occur due to incorrect location recall by the UKB participant, or due to differential recording by interviewers (e.g., recorded at different levels of detail). "

I'd argue there is a fourth reason: UK Biobank birthplaces are recorded to a list of 43,000 placenames, specifically towns or districts (which is described here: <https://biobank.ndph.ox.ac.uk/showcase/refer.cgi?id=100235>) which exist in a tree-like structure and if not matched the birthplace could be freely entered. Then these locations were mapped to a 1 by 1 km grid (see here for details: <https://biobank.ndph.ox.ac.uk/showcase/refer.cgi?id=118122>). If one of two siblings answered "London" while the other answered "Peckham" (which is in London), these would likely be mapped to a different grid coordinate, but it wouldn't be a response error, nor a transcription error on the part of the interviewer.

So, I see 4 sources of difference:

1. Actual move
2. Reporting error
3. Interviewer error
4. Error in processing from reported birthplace to grid coordinates

I think it would be good to outline very explicitly these as separate in the intro, if you wish you can then state that 2-4 would result in error and can therefore in terms of their consequences be grouped.

Issue 2, ignoring the repeated measures:

There are up to ~28,000 ppl who reported their birthplace at least twice in the second visit and imaging visit.

With minimal assumptions you can use these to independently infer error and significantly strengthen your conclusions? These cannot have occurred due to an actual move, you can argue that chance they'd have the same interviewer are slim (and become smaller if the time between interviews is longer, and almost 0 when people visit different centres) which would allow you to separate out interviewer from reporting error

It like to at least see the error in repeated birthplace reports by the same individual compared to the error implied by siblings.

Issue 3, lack of common basis for the reporting of "region" might conflate with centered specific interviewer error.

It is my understanding UK towns can be classified into several types of subdivisions, some are functional and well known (NHS trusts, postal regions, city limits, electoral constituencies) some are technical and specifically for researchers or policy (MSOA regions etc). The authors report a birthplace error rate per UKBiobank assessment center and point to differences in interviewer error rate as a potential cause.

I'd argue these differences might simply arise because certain areas underwent more growth & renaming (towns being absorbed into London etc.), or redistricting, causing the same location to become known under multiple names (e.g. "Battersea", "Wandsworth" and "London" refer to the same place, but Wandsworth borough didn't exist until 1965). These changes can result in "errors" in various steps of birthplace transcription to the data coded into a latitude/longitude grid as available from UKB.

I propose the authors compute the error rate per region, and map it, as it might reveal insight as to whether assessment center is a reasonable cause of the error rate difference or whether regions that underwent steep changes in social change have higher error rates.

Issue 4, authors mapping of grid coordinates.

Authors map the 1km grid coordinates to parishes, districts and counties as defined in 1951. It would be far better to map to the original UKB tree of regions as presented to the interviewer. It would be superior as it's a tree structure and so it would be very easy to determine whether the various transformations from birthplace to a grid (by UKB) and from a grid to birthplace (by the authors) cause further error. I am not saying the authors did anything wrong, we all must take this second step in any type of analysis with the data, but the step does potentially cause further errors.

As far as I can tell this list of places isn't publicly available, I urge to authors to either:

1. As for this list from UKB and do sensitivity analysis on whether the error rate is higher/lower
2. If they can't get it in a reasonable time (days/weeks) its might be possible simply reflect on the utility of access to this list and its possible effects on error rate where UKB to release it in the future.

Issue 5, a further statistical design affected.

The authors explore attenuation bias when a variable of interest (say noise pollution or regional employment) is falsely assigned due to error in birthplace. However, I urge them to also consider the effects on analysis that use birthplace as a fixed effect (or random but since the authors are economists, fixed would likely be preferred by them).

For example, this recent paper: Abdellaoui, A., Dolan, C. V., Verweij, K. J., & Nivard, M. G. (2022).

Gene–environment correlations across geographic regions affect genome-wide association studies. *Nature genetics*, 54(9), 1345-1354. Uses birthplace as a fixed effects covariate and assesses its effect on GWAS. The authors do remark differential error in birthplace vs current address recording in UKB might affect results, but they forgo any analyses.

It could be that I am thinking too lightly on the assumptions needed, and scope of, for further simulation where birthplace is a fixed effect and if that's the case, I am happy for the authors to persuade me they wouldn't be appropriate for this article. Currently they simply remark:

"The consequences of this measurement error therefore depend on what the birth location data are used for. "

I'd like a clearer rationale of why this specific focus, and perhaps a list of other types of analyses (like for example fixed effects correction for birthplace) the authors view as potentially affected by birthplace error. This could help others assess the risk to their own analyses better.

Issue 5, the PCA analysis

1. Figure 4 needs standard errors, is there a decrease with distance? I don't know?

2. both a real move, and an error would induce a reduction in correlation with the PCA of your neighbors, but a real move would more likely affect the second sib (born after the move) while an error could affect either? So, what if you split these plots into oldest and youngest? The paralleled decay would be error and differences in decay couldn't really be error. Alternatively plot age difference between sibs against the PC correlation to test that's flat(er)?

3. As the PC's are orthogonal, and the pattern isn't extremely clear as is (but maybe it is? Let's see those standard errors...), perhaps do joint analysis of the 5PCs? The MSE from the region mean PC can be summed across PC1 to PC5 as their orthogonal, would this be a more powerful indicator?

Reviewer #2 (Remarks to the Author):

von Hinke and Vitt analyze twin and sibling retrospective birth location data from the UKB and find that these data are often inaccurate. They show, through simulations, that under some assumptions and under some conditions, this can introduce large amounts of attenuation bias in analyses that use such birth location data to estimate the effects of a location-specific exposure on an outcome.

The authors have conducted a nice array of detailed analyses and the paper is well written (though I have some comments regarding organization and emphasis, below). However, although the authors make an important point, I'm not sure the contribution is significant enough for *Nature Communications* (I am on the fence here). Below are some more specific comments.

1. To help increase the significance of the contribution, it'd be very helpful if the authors could document and discuss a number of high-profile studies that used retrospective birth location data from the UKB (as opposed to just citing some such studies in passing in the intro paragraph). Ideally, they could replicate the analysis of at least one such study that did not use sibling FEs, and show how their suggestion of only using data from sibs born in the same location (for analyses without sib FEs) impacts the results. In other words, it'd be very helpful if the authors provided some real examples of analyses that have been conducted and that can be fixed.

2. The paper makes 3 distinct contributions. First, it analyzes data on twins and sibs and finds that there are important inaccuracies in retrospective birth location data in the UKB. Second, it shows that

this leads to attenuation bias. And third, buried in the Discussion section, the authors discuss some possible solutions, including some solutions that are unlikely to work. This third contribution is important but is not prominent enough in the paper. I think the authors should have a section titled "Potential solutions" (or something like that) with separate subsections for the different potential solutions and with a discussion of each potential solution. Much of that text is already in the Discussion section, but the authors could provide some more details, as this material will include (in my opinion) the main takeaways readers will remember from the paper. The authors should also mention this contribution in the abstract and in the intro. And if possible, as per the previous point, include one or more examples where they apply a workable solution.

3. Relatedly, there is a disconnect between the simulations and discussion in the attenuation bias part of the paper---where it is assumed that classical measurement error is present---and the discussion of ORIV, where the authors state that measurement error is not classical. The authors discuss the fact that measurement error is unlikely to be classical in the Attenuation bias section of the Methods, but this should be mentioned clearly in the main text. And the authors should discuss in the main text how far we are likely to be from the classical measurement error assumption. This is important, because if measurement error is not far from being classical, then in many studies it may be just fine to use retrospective birth location data without care to the fact that it contains error, as long it is emphasized that there is measurement error in the data and that this will attenuate the results downwards (in magnitude). And if measurement error is far from being classical, then what is one to make of the authors' simulations and discussion of attenuation bias, which assume classical measurement error?

4. The authors should highlight in the main text some of the key assumptions they make. Currently, these are listed in the Methods only, but they are critical for a good understanding of what the authors are doing and so should be at least briefly mentioned and discussed in the main text. (I had many questions while reading the paper and found answers to most of them only when I got to the Discussion and Methods sections; readers should be given heads up that these important issues are discussed there.)

The remaining comments are more minor (but most should nonetheless be addressed):

5. On p. 5, the following phrase confused me:

"The bias is increasing in the variance of the measurement error, which is driven by ... (iii) by the share of (non-spatial) variation over time exhibited by the environmental measure."

I was able to understand what the authors mean by reading further, but the authors should rephrase this point (iii) (e.g., "the share of the variation in the exposure that is driven by temporal (vs. spatial) variation")

6. In the simulations for the attenuation bias section, there are a few things I couldn't understand (even after closely reading the Methods section):

i. Why simulate 10 district-level variables (vs. one)?

ii. How is the exposure defined/constructed exactly --- i.e., how do the authors get from the V 's to the X ?

iii. What does ρ capture, exactly, in plain English? I understand that it's the spatial autocorrelation parameter, but how is it scaled? For instance, what does a ρ of 0.05 vs. 0.25 imply, exactly? Is it Moran's I , or what metric is it based on?

iv. In formula (12), ϵ is standard normal, but that does not mean much unless we know what's the scale (or the variance) of X ? Does that vary across X 's, or do you standardize X ? If the former, then it's hard to compare the results in Figure 3.

RESPONSE TO REFEREE 1

(Referee's original comments in italics, our responses in bold, page numbers refer to the manuscript version without tracked changes)

Prof. Von Hinke & Dr. Vitt present what can only be described as an essential public evaluation of the quality of birthplace reports in UK Biobank, and to an extent of retroactively reported birthplace in general. I have a series of what I feel are important comments and remarks but in general the value of the work is obvious and the quality high.

I have a lot of comments because I kind of love this article and see its utility to the field, I urge the editor not to mistake the quantity of feedback for a negative overall assessment.

Thank you for your encouraging words and for your detailed comments and suggestions on our previous submission. These have been of great help in revising our paper. We explain how we addressed each of your points in turn below (in bold).

Issue 1, taxonomy:

The authors study the frequently with which birthplaces are misreported, in their intro they go on to state that there are two causes why siblings could have different birthplaces:

“Assuming that the siblings grew up together, their location of birth can differ for two reasons. First, the family may have moved house between the births of their two or more children (i.e., a ‘true’ change in their birth location). Second, the location of birth may have been misreported by one or more children (i.e., an ‘error’ in their birth location).”

The authors then say that in fact the second reason is a composite of 2 reasons:

“The latter can occur due to incorrect location recall by the UKB participant, or due to differential recording by interviewers (e.g., recorded at different levels of detail).”

I'd argue there is a fourth reason: UK Biobank birthplaces are recorded to a list of 43,000 placenames, specifically towns or districts (which is described here: <https://biobank.ndph.ox.ac.uk/showcase/refer.cgi?id=100235>) which exist in a tree-like structure and if not matched the birthplace could be freely entered. Then these locations were mapped to a 1 by 1 km grid (see here for details: <https://biobank.ndph.ox.ac.uk/showcase/refer.cgi?id=118122>). If one of two siblings answered “London” while the other answered “Peckham” (which is in London), these would likely be mapped to a different grid coordinate, but it wouldn't be a response error, nor a transcription error on the part of the interviewer.

So, I see 4 sources of difference:

- 1. Actual move*
- 2. Reporting error*
- 3. Interviewer error*
- 4. Error in processing from reported birthplace to grid coordinates*

I think it would be good to outline very explicitly these as separate in the intro, if you wish you can then state that 2-4 would result in error and can therefore in terms of their consequences be grouped.

Thank you for this, that's a good point. We have now added “error in processing” as a fourth source of location difference between siblings on p.2, where we say:

“Assuming that the siblings grew up together, their location of birth can differ for two reasons. First, the family may have moved house between the births of their two or more children (i.e., a ‘true’ change in their birth location). Second, there may be an error in the birth location recorded for (at least) one of the siblings. For the former, we assume that a longer spacing between births linearly increases the probability of a house move; something we test empirically below. The latter can occur due three reasons. First, a UKB participant could have incorrectly recalled their location of birth. Second, any differential recording across interviewers can cause error in the location (e.g., recorded at different levels of detail). Third, it may have been caused by errors in processing the reported birthplace to grid coordinates if siblings differ in the precision of their reporting (holding constant the interviewer). We refer to the latter three ‘location-errors’ as measurement error.”

Issue 2, ignoring the repeated measures:

There are up to ~28,000 ppl who reported their birthplace at least twice in the second visit and imaging visit.

With minimal assumptions you can use these to independently infer error and significantly strengthen your conclusions? These cannot have occurred due to an actual move, you can argue that chance they’d have the same interviewer are slim (and become smaller if the time between interviews is longer, and almost 0 when people visit different centres) which would allow you to separate out interviewer from reporting error

It like to at least see the error in repeated birthplace reports by the same individual compared to the error implied by siblings.

This is a great additional source of variation that we were not aware of – thank you for highlighting this. We have added a supplementary analysis based on these repeated measures and discuss its results on p.5/6. We here highlight that this approach avoids any variation that is driven by house moves. Instead, the error probability in the repeated measures sample captures a combination of interviewer error (as interviewers are likely to have differed between interviews) as well as *time-varying* individual reporting error, the latter of which may also lead to some processing error. In other words, this analysis allows us to distinguish between variation driven by house moves and measurement error, but we cannot cleanly separate out the latter into interviewer, reporting and processing error. We say:

“In Appendix E we conduct a supplementary analysis of the measurement error in birth location using a UKB sub-sample of participants who were asked to report their birth location more than once. This allows us to distinguish between variation driven by house moves (which can be ruled out in repeated reports of birth location) and measurement error (which cannot). Similar to the sibling approach, the error probability estimated in this sub-sample captures interviewer effects, since it is likely that interviewers differed between siblings and between repeat interviews. Additionally, the repeat measures will also capture *time-varying* individual-reporting error, which in turn may also lead to processing error. *Time-invariant* individual-reporting error, however, will not be captured in the analysis of repeat measures.

Our findings show that the error probabilities in participants' birth location that are based on the repeat measures are lower than those based on the sibling comparison in Table 1. For small error distances up to 5km as well as for parish and district errors the difference in probabilities to the sibling comparison are relatively small (e.g. 24.6% vs 28.4% for any error distance). However, for larger error distances the error probabilities derived from the repeat location reports are substantially lower (e.g. 1.2% vs 3.4% for error distances over 50km). Thus, our results suggest that short distance errors are mostly driven by interviewer, time-varying individual reporting and processing effects (which are captured to a similar degree by both approaches). Long distance errors on the other hand, are more likely to be driven by time-invariant individual reporting errors (which are only captured in the sibling comparisons)."

The corresponding methods are described on p.24/25 (not copied here to ensure brevity of our reply) and the results are shown in the new Appendix E.

Issue 3, lack of common basis for the reporting of "region" might conflate with centered specific interviewer error.

It is my understanding UK towns can be classified into several types of subdivisions, some are functional and well known (NHS trusts, postal regions, city limits, electoral constituencies) some are technical and specifically for researchers or policy (MSOA regions etc). The authors report a birthplace error rate per UKBiobank assessment center and point to differences in interviewer error rate as a potential cause.

I'd argue these differences might simply arise because certain areas underwent more growth & renaming (towns being absorbed into London etc.), or redistricting, causing the same location to become known under multiple names (e.g. "Battersea", "Wandsworth" and "London" refer to the same place, but Wandsworth borough didn't exist until 1965). These changes can result in "errors" in various steps of birthplace transcription to the data coded into a latitude/longitude grid as available from UKB.

I propose the authors compute the error rate per region, and map it, as it might reveal insight as to whether assessment center is a reasonable cause of the error rate difference or whether regions that underwent steep changes in social change have higher error rates.

This is a good point. We agree that variation in interviewer error rates is not the only cause of the variation in error rates by assessment centres. Indeed, the frequency of renaming/boundary redrawing – as you mention – is another. Another one again is the size of the areas on the list used by interviewers. For example, regions that include many small areas are likely to have a higher error rate than regions with fewer larger areas. Both interviewer and processing errors are less likely to occur in regions with fewer large areas. Similarly, small distance reporting errors (and short distance moves) are less likely to show if areas are large.

As you suggest, in addition to reporting the error rates by assessment centre, we have now computed the error rates by region of birth. Although the findings are consistent with each other – for example, we find some of the lowest error rates for the Edinburgh assessment centre as well as for Scotland as a whole – we note it is not possible to distinguish between whether the error is driven by interviewers, boundary changes, or other plausible explanations.

We now discuss these issues on p.34 (Appendix D), where we say:

“In addition to characteristics of the UKB participants and their area of birth, measurement error may also be related to the UKB assessment centre at which they were first interviewed. The birth location data is based on an interaction between the participant and an interviewer, who identified the place of birth from a long list of place names in the computer system using the information given by the participant. Thus the accuracy of the recorded birth location may be subject to interviewer effects. Similarly, any errors in processing reported locations of birth to grid coordinates by the data providers may cause measurement error. For example, changes in regions’ boundaries may lead one sibling to report a different location from the other sibling as their area of birth was known under a different name, even though they did not move. Hence, variation in measurement error by assessment centre or region of birth may be driven by interviewer effects or processing errors. Although we cannot distinguish between the two, we explore their (joint) importance by studying heterogeneity in the error probability across the 22 assessment centres of the UKB as well as the 12 regions of birth that we observe. Panel (a) of Figure D.1 shows substantial heterogeneity in the discordance share for siblings’ district of birth: only 26% of participants interviewed in Edinburgh are recorded to have a different district of birth to their sibling, whilst this is 55% of those interviewed in Barts. Naturally, these differences translate to substantial heterogeneity in the estimated probability of measurement error shown in panel (b), from 11% in Edinburgh to 30% in Barts. In line with this, we find a low error probability for siblings born in Scotland and one of the highest in the London area (see Figure D.2).”

Because of the many different potential causes of variation in error rates, we argue that it is impossible to distinguish between these multiple explanations. However, we do explore some of these indirectly here. To shed further light on the relevance of boundary changes and the size of areas, we have conducted an additional supplementary analysis. We repeat our main estimations (Table 1) with two added control variables: (1) the number of boundary changes to the reported district of birth between 1940 and 1973 and (2) the area (in km²) of the reported parish of birth, capturing its size. The results show no consistent impact of an increase in the number of boundary changes on the discordance in siblings’ birth locations. A larger size of the local area, however, is found to significantly reduce the discordance share. We do not include this additional analysis in the paper to maintain the focus on the heterogeneity in error rates, but show the results for your information below.

Table R.1: Differences in siblings' birth location and their age gap - controlling for area characteristics

	Different birth location:							
	(1) Parish	(2) District	(3) County	(4) d>0km	(5) d>5km	(6) d>10km	(7) d>30km	(8) d>50km
Age gap (years)	0.009*** (0.001)	0.009*** (0.001)	0.007*** (0.001)	0.012*** (0.001)	0.010*** (0.001)	0.007*** (0.001)	0.007*** (0.001)	0.006*** (0.001)
Parish area (square km)	-0.002*** (0.000)	-0.002*** (0.000)	-0.001*** (0.000)	0.000*** (0.000)	-0.001*** (0.000)	-0.001*** (0.000)	-0.001*** (0.000)	-0.000*** (0.000)
No. district changes 1940-73	-0.005 (0.005)	-0.005 (0.005)	0.011*** (0.004)	0.004 (0.005)	0.003 (0.005)	-0.004 (0.004)	-0.009*** (0.003)	-0.004 (0.003)
Constant	0.427*** (0.008)	0.405*** (0.008)	0.157*** (0.006)	0.470*** (0.008)	0.290*** (0.007)	0.216*** (0.006)	0.125*** (0.005)	0.094*** (0.005)
Derived probabilities:								
\hat{q} (move probability)	0.009*** (0.001)	0.009*** (0.001)	0.007*** (0.001)	0.012*** (0.001)	0.010*** (0.001)	0.007*** (0.001)	0.007*** (0.001)	0.006*** (0.001)
\hat{p} (error probability)	0.168*** (0.004)	0.161*** (0.004)	0.055*** (0.003)	0.285*** (0.005)	0.134*** (0.004)	0.083*** (0.003)	0.045*** (0.002)	0.033*** (0.002)
N (sibling pairs)	16,910	16,910	16,910	16,910	16,910	16,910	16,910	16,910

Note: The sample is restricted to the first sibling pair observed in each family. The control variables are characteristics of the local area of birth reported by the first (i.e. older) sibling: the area size of their reported parish of birth (in km^2) and the number of boundary changes for their reported district of birth between 1940 and 1973). The estimated error probability \hat{p} was computed at the average parish area size and for a district without boundary changes. Heteroskedasticity robust standard errors are shown in parentheses. Standard errors for \hat{p} were computed using the delta method. Significance levels are indicated as follows: * $p < 0.1$, ** $p < 0.05$, *** $p < 0.01$

Issue 4, authors mapping of grid coordinates.

Authors map the 1km grid coordinates to parishes, districts and counties as defined in 1951. It would be far better to map to the original UKB tree of regions as presented to the interviewer. It would be superior as it's a tree structure and so it would be very easy to determine whether the various transformations from birthplace to a grid (by UKB) and from a grid to birthplace (by the authors) cause further error. I am not saying the authors did anything wrong, we all must take this second step in any type of analysis with the data, but the step does potentially cause further errors.

As far as I can tell this list of places isn't publicly available, I urge to authors to either:

1. *As for this list from UKB and do sensitivity analysis on whether the error rate is higher/lower*
2. *If they can't get it in a reasonable time (days/weeks) its might be possible simply reflect on the utility of access to this list and its possible effects on error rate where UKB to release it in the future.*

Thank you for this suggestion. Our main reason for mapping the 1km grid coordinates to parishes, districts and counties (as defined in 1951) is that a substantial amount of historic data is available at these levels of aggregation, rather than at the UKB tree of regions. Our aim is therefore to explore the error rates at these aggregation levels and to quantify the size of the corresponding attenuation bias, which is particularly relevant when using data at these levels of aggregation.

Following your suggestion however, we have contacted the UK Biobank and they have provided the list of locations in England and Wales used in the interview as well as their corresponding coordinates. This allows us to illustrate how the answers recorded by the interviewer are mapped to grid coordinates. We have added two maps of birth locations used in the interviews and their corresponding 1km grid coordinates for examples of an urban and a rural area in Appendix B.

This shows that “reverse mapping” from the rounded birth coordinates (denoted by the ×’s) to the locations provided in the interview (denoted by the blue dots) is not possible since in many cases multiple locations are mapped to the same grid coordinates (especially in urban areas). For many applications, however, the benefits of “reverse mapping” are unclear. Firstly, the list locations are not used in applications of the birth location data (e.g. to merge external data or as control variables). Secondly, our current analysis examines both the raw grid coordinates and the parish/district/county, and therefore is able to show that even prior to any mapping from the grid coordinates to aggregated areas there is substantial error. We now highlight these points on p.15 where we say:

“We focus on birth location data in the form of 1km grid coordinates as well as historical parishes, districts, and counties since most applications use information aggregated to these geographical identifiers (e.g. to merge external data or as control variables). Furthermore, analysing the actual grid coordinates allows us to examine errors before the mapping to historical areas may introduce additional processing errors.”

Issue 5, a further statistical design affected.

The authors explore attenuation bias when a variable of interest (say noise pollution or regional employment) is falsely assigned due to error in birthplace. However, I urge them to also consider the effects on analysis that use birthplace as a fixed effect (or random but since the authors are economists, fixed would likely be preferred by them).

For example, this recent paper: Abdellaoui, A., Dolan, C. V., Verweij, K. J., & Nivard, M. G. (2022). Gene–environment correlations across geographic regions affect genome-wide association studies. *Nature genetics*, 54(9), 1345-1354. Uses birthplace as a fixed effects covariate and assesses its effect on GWAS. The authors do remark differential error in birthplace vs current address recording in UKB might affect results, but they forgo any analyses.

It could be that I am thinking to lightly on the assumptions needed, and scope of, for further simulation where birthplace is a fixed effect and if that’s the case, I am happy for the authors to persuade me they wouldn’t be appropriate for this article. Currently they simply remark:

“The consequences of this measurement error therefore depend on what the birth location data are used for.”

I’d like a clearer rationale of why this specific focus, and perhaps a list of other types of analyses (like for example fixed effects correction for birthplace) the authors view as potentially affected by birthplace error. This could help others assess the risk to their own analyses better.

Thank you, this is a really interesting point. Our focus on the attenuation bias that occurs when using a birthplace-based variable of interest was motivated by the growing number of research applications following this approach (including our own). We agree that birthplace fixed effects are another common application. We have therefore conducted additional simulations to assess the bias when area of birth fixed effects are subject to measurement error. The methods are described on pages 22-24 (not copied here to ensure brevity of our reply) and the results are included in the new Appendix G. On pages 10/11 we describe the results of these additional simulations as follows:

“Even if the variable of interest is measured *without* error, the corresponding regression coefficient may be subject to omitted variable bias when control variables (including birth-

location fixed effects) are subject to measurement error, thereby omitting part of the “true” control variable^{29–31}. In other words, measurement error in the birth location data may lead to bias in the coefficient of interest in analyses that rely on birth-location fixed effects as control variables³². This bias differs in important aspects from the attenuation bias when the *variable of interest* (i.e., not the control variable) is measured with error. In the case of classical measurement error in a control variable, the resulting “partially omitted variable bias” in the variable of interest will be smaller in magnitude and in the same direction as the bias from fully omitting the control variable. Thus, measurement error of control variables will not necessarily cause attenuation bias in the coefficient of interest, but indeed can lead to bias away from zero. Unlike classical attenuation bias, the “partially omitted variable bias” is not proportional to the coefficient of interest and can therefore arise even when the variable of interest has no impact on the outcome.

We use Monte Carlo simulations to quantify the size of this bias in estimations controlling for district of birth fixed effects when the variable of interest is correctly observed but the district of birth is subject to measurement error. Figure G.1 shows the size of the bias for different spatial autocorrelations (ρ) in the district fixed effects and different correlations between the fixed effects and the variable of interest. Our simulations show the bias to be proportional to the ratio of the standard deviation of the district fixed effects to that of the variable of interest (σ_μ/σ_X). We therefore pool the simulation results for different standard deviations and express the bias in units of σ_μ/σ_X .

The direction of the bias corresponds to the sign of the correlation between the variable of interest and the fixed effects: if they are positively (negatively) correlated, the bias is positive (negative). Similar to the bias from fully omitted control variables, the magnitude of the bias is increasing in the correlation of the variable of interest and the control variable, in this case the district fixed effects. A higher spatial autocorrelation in the district fixed effects reduces the size of the bias, with errors in the district of birth having smaller consequences for the fixed effects.

Hence, in summary, our simulations show that measurement error in birth locations may lead to substantial bias in regressions that control for place of birth fixed effects, even when the variable of interest is measured accurately. The bias will be particularly large in the presence of substantial fixed effects relative to the variation in the variable of interest (i.e., a large value for σ_μ/σ_X), when the variable of interest is strongly correlated with the fixed effects and when there is little spatial correlation in the fixed effects.”

Issue 5, the PCA analysis

1. Figure 4 needs standard errors, is there a decrease with distance? I don't know?

You are right that the figure should have shown standard errors. We have now replaced the graph with one that additionally reports the 95% confidence intervals, showing that the correlation decreases significantly with an increase in the distance to siblings' birth locations. We mention this on p.11, saying:

“The vertical axis of Figure 4 shows the correlation of each genetic principal component between individuals in the sibling sample and the mean among individuals in the non-sibling sample who reported the same birth location, including their 95% error bars. We estimate these correlations separately for siblings

with different levels of discordance in their reported birth location, measured along the horizontal axis. Correlations between siblings without any discordance in birth location and others reporting the same birth location are above 0.35 for the first five principal components, with some as high as 0.6. As the distance between siblings' reported birth location increases in Figure 4, the correlation with others reporting the same birth location reduces. Indeed, when comparing siblings who reported being born more than 200 kilometres apart (3.6% of sibling pairs) to those without any discordance, the correlation coefficients decrease by more than 40% for all five spatially structured principal components. These are significant differences for each of the first five principal components. (...) These results illustrate the impact of household mobility and measurement error on the estimation of the spatial structure of genetic data."

2. both a real move, and an error would induce a reduction in correlation with the PCA of your neighbors, but a real move would more likely affect the second sib (born after the move) while an error could affect either? So, what if you split these plots into oldest and youngest? The paralleled decay would be error and differences in decay couldn't really be error. Alternatively plot age difference between sibs against the PC correlation to test that's flat(er)?

This is a good suggestion. We had in fact done the analysis separately by first and laterborns, but this did not show any differential correlations, which is why we didn't report this in our initial submission. We now show this below in figure R.1 and report this on p.11 of the paper (we do not show the figure in the paper), where we say:

"Note that the figures look similar when we separately plot them for first- and laterborns."

Figure R.1: Correlation of siblings' genetic principal components with others reporting the same birth location - by birth order

Note: The figures show the correlation between genetic principal components of first and second born individuals in our sibling sample with the mean genetic principal components of non-sibling UKB participants who reported the same birth coordinates. Correlations are shown separately for the first 6 genetic principal components (PC1 - PC6) and for distances between the reported birth locations of the individuals and their siblings. Principal components are based on principal component analysis conducted on unrelated white-british individuals in the UKB. SNPs were filtered based on minor allele frequency > 0.01 and clumped for linkage disequilibrium based on minor allele frequency ($R^2 > 0.1$). Long-range linkage disequilibrium regions were removed. Vertical bars represent 95% confidence intervals.

3. As the PC's are orthogonal, and the pattern isn't extremely clear as is (but maybe it is? Let's see those standard errors...), perhaps do joint analysis of the 5PCs? The MSE from the region mean PC can be summed across PC1 to PC5 as their orthogonal, would this be a more powerful indicator?

Thank you. Since the individual confidence intervals added based on your comment above show a clear pattern, we decided against conducting a joint analysis. We hope you agree that our results above reduce the importance of such additional analysis.

RESPONSE TO REFEREE 2

(Referee's original comments in italics, our responses in bold, page numbers refer to the manuscript version without tracked changes)

von Hinke and Vitt analyze twin and sibling retrospective birth location data from the UKB and find that these data are often inaccurate. The show, through simulations, that under some assumptions and under some conditions, this can introduce large amounts of attenuation bias in analyses that use such birth location data to estimate the effects of a location-specific exposure on an outcome.

The authors have conducted a nice array of detailed analyses and the paper is well written (though I have some comments regarding organization and emphasis, below). However, although the authors make an important point, I'm not sure the contribution is significant enough for Nature Communications (I am on the fence here). Below are some more specific comments.

Thank you for your detailed comments and suggestions on our previous submission, which have been of great help in revising our paper. We explain how we addressed each of your points in turn below (in bold).

1. To help increase the significance of the contribution, it'd be very helpful if the authors could document and discuss a number of high-profile studies that used retrospective birth location data from the UKB (as opposed to just citing some such studies in passing in the intro paragraph). Ideally, they could replicate the analysis of at least one such study that did not use sibling Fes, and show how their suggestion of only using data from sibs born in the same location (for analyses without sib Fes) impacts the results. In other words, it'd be very helpful if the authors provided some real examples of analyses that have been conducted and that can be fixed.

Thank you. We have now written a more detailed discussion of a selection of high-profile papers that use retrospective birth location data in their analyses. We have decided to put this in a new Appendix A to avoid disrupting the flow of the paper but refer the reader to this Appendix on p.1.

Regarding your second suggestion of replicating a study to show the importance of exploring the robustness of the results to only using data from siblings that are born in the same location, we have replicated the phenotype comparisons from Abdellaoui et al. (2019). More specifically, Appendix I shows the differences in a range of phenotypes (years of education, body fat, height, BMI and self-rated overall health) between individuals born in coal mining and non-coal mining areas, as well as between four migration groups defined based on the coal-mining status of individuals' place of birth and current place of residence (i.e., those who moved from a coal mining area to a non-coal area; those who stayed in a non-coal area; those who moved from a non-coal area to coal mining area; and those who stayed in a coal mining area). We conduct these comparisons on two sub-samples: the sample of sibling pairs used in our main analysis (i.e., restricted to the two oldest siblings per family) and a sample with more "robust" birth location data ("robust sibs"; dropping sibling pairs that differed in the coal mining status of their place of birth). For the "robust" sub-sample we match siblings' location of birth at the geographical level of variation used; in this case coal versus non-coal areas. Hence, the "robust" sample includes siblings that either both report being born in a coal area, or both report not being born in a coal area. These analyses show that the phenotypes in our two sub-samples differ, with the "robust" sub-sample showing more variation between those born in coal and non-coal areas. Not taking this into account underestimates those phenotypic differences. We

refer to this Appendix in the new “Possible solutions” section of the paper (see your next comment).

2. The paper makes 3 distinct contributions. First, it analyzes data on twins and sibs and finds that there is important inaccuracies in retrospective birth location data in the UKB. Second, it shows that this leads to attenuation bias. And third, buried in the Discussion section, the authors discuss some possible solutions, including some solutions that are unlikely to work. This third contribution is important but is not prominent enough in the paper. I think the authors should have a section titled “Potential solutions” (or something like that) with separate subsections for the different potential solutions and with a discussion of each potential solution. Much of that text is already in the Discussion section, but the authors could provide some more details, as this material will include (in my opinion) the main takeaways readers will remember from the paper. The authors should also mention this contribution in the abstract and in the intro. And if possible, as per the previous point, include one or more examples where they apply a workable solution.

Thank you for this suggestion. In response, we have restructured the results and conclusion sections. We have moved our discussion of the potential solutions to the inaccuracies in reported retrospective birth location data in a separate subheading in the results section (though we have not used additional sub-subheadings for each solution), highlighting advantages as well as potential drawbacks of each suggestion. We do not copy our discussion here as it is almost a full page but refer you to p.11-13 of the paper. As you suggest in your comment, the discussion now emphasizes our main contributions and summarizes our results. We have also rewritten the abstract and introduction to reflect these three distinct contributions.

3. Relatedly, there is a disconnect between the simulations and discussion in the attenuation bias part of the paper---where it is assumed that classical measurement error is present---and the discussion of ORIV, where the authors state that measurement error is not classical. The authors discuss the fact that measurement error is unlikely to be classical in the Attenuation bias section of the Methods, but this should be mentioned clearly in the main text. And the authors should discuss in the main text how far we are likely to be from the classical measurement error assumption. This is important, because if measurement error is not far from being classical, then in many studies it may be just fine to use retrospective birth location data without care to the fact that it contains error, as long it is emphasized that there is measurement error in the data and that this will attenuate the results downwards (in magnitude). And if measurement error is far from being classical, then what is one to make of the authors’ simulations and discussion of attenuation bias, which assume classical measurement error?

Thank you for bringing this up. It highlights that the discussion of our simulations was not sufficiently clear. Indeed, we do *not* assume classical measurement error in our simulations of birth location errors. We discussed the classical measurement error case only to illustrate the general issue of attenuation bias but do not use this further. To avoid any confusion, we have therefore decided to move this discussion to (the new) Appendix K and clarify early (p.2) on that we do *not* assume classical measurement error in our simulations.

Following your suggestion however, we have attempted to quantify how far the measurement error in our setting is from classical measurement error. The major difference from normally

distributed measurement error is that a large share of observations has a measurement error of zero, the share of not mismeasured birth locations is $1-p$. For district-level data, this is approximately 84% based on our estimates in Table 1. In Appendix L (Figure L.1), we plot the distribution of non-zero measurement errors for our simulated district-level data and observe further deviations from normality in the case of large spatial correlation in the variable of interest: for small spatial correlations, the non-zero measurement errors are approximately normally distributed. However, as the spatial correlation increases, the kurtosis of the error distribution increases leading to a leptokurtic distribution.

A further assumption of classical measurement error is that the measurement error is uncorrelated with the true (unobserved) variable. In most circumstances, this assumption will fail due to regression to the mean and ceiling effects – an observation with a high (low) level for the true variable is more likely to have a mismeasured level below (above) the true one. Again, we quantify the deviation from this assumption using our simulated district-level data in Appendix L (Tables L.1 and L.2). We observe large negative correlations between the measurement error and the true variable, in particular when focusing on observations with non-zero measurement error.

We discuss the above in the *Methods* section on p.18-19, where we say:

“Our application is unlikely to show classical measurement error. For one, with the majority of siblings reporting the same birth location, there is a spike at zero. Even when ignoring the zeros, normality of the measurement errors may not hold in our setting. In Figure L.1 of Appendix L we show that the distribution of errors in our district-level simulations is approximately normal when spatial autocorrelations are small. Large levels of spatial correlation, however, can lead to a leptokurtic distribution. Furthermore, one would expect a negative correlation between the true explanatory variable and the measurement error: if an exposure is high (low) for the true birth location, it is more likely that the exposure in the misreported birth location is below (above) the true exposure level due to regression to the mean. We confirm this expectation in Table L.1 and Table L.2 of Appendix L, which show large negative correlations between the true explanatory variable and the measurement error in our district-level simulations; in particular when focusing on observations with non-zero measurement error.

(...)

The simulations are based on the birth location differences observed in the data, and therefore do not assume classical measurement error.”

4. The authors should highlight in the main text some of the key assumptions they make. Currently, these are listed in the Methods only, but they are critical for a good understanding of what the authors are doing and so should be at least briefly mentioned and discussed in the main text. (I had many questions while reading the paper and found answers to most of them only when I got to the Discussion and Methods sections; readers should be given heads up that these important issues are discussed there.)

Thank you for this suggestion. We have added the following (brief) description of the key assumptions made to derive the error and move probabilities (see p.4/5):

“These derivations rely on a number of simplifying assumptions: (1) if siblings report the same birth location, we assume this is the true birth location, (2) biological siblings have grown up in the same household, (3) the move probability increases linearly with the age gap between two siblings, (4) errors in the birth location occur randomly with the same probability across all participants, are independent within sibling pairs and independent of the sibling age gap.”

We then refer the reader to the *Methods* section for a more detailed discussion of these assumptions.

The remaining comments are more minor (but most should nonetheless be addressed):

5. On p. 5, the following phrase confused me:
“The bias is increasing in the variance of the measurement error, which is driven by ... (iii) by the share of (non-spatial) variation over time exhibited by the environmental measure.”
I was able to understand what the authors mean by reading further, but the authors should rephrase this point (iii) (e.g., “the share of the variation in the exposure that is driven by temporal (vs. spatial) variation”)

Thank you, your suggested phrase is indeed much clearer. We have rephrased the sentence (p.6 in the revised paper) as follows:

“The bias is increasing in the variance of the measurement error, which is driven (i) by the probability of an error in the birth location, (ii) by the difference in the exposure between the true and the reported place of birth, and (iii) by the share of the variation in the exposure that is driven by temporal (vs spatial) variation.”

We use the same words in Appendix K when we discuss the case of classical measurement error.

6. In the simulations for the attenuation bias section, there are a few things I couldn’t understand (even after closely reading the Methods section):
 - i. Why simulate 10 district-level variables (vs. one)?

Thank you. We simulate 10 district/parish/coordinate-level variables to ensure that our simulation results are not driven by an “outlier” in the spatial simulations. Especially when the number of spatial units is relatively small, a single simulated variable might not be a good representation of the spatial correlation targeted. We now mention this on p.19.

“We simulate 10 variables (rather than 1) to ensure that our simulation results are not driven by an “outlier” in the spatial simulations, which is particularly important when the number of spatial units is relatively small.”

- ii. How is the exposure defined/constructed exactly --- i.e., how do the authors get from the V’s to the X?

Thank you. In our description of the bias simulations, we denote simulated time-varying spatially correlated variables as $V_{a,\rho,k}$. These are variables defined at the area-by-year level, which have not yet been merged to the individual-level data.

In order to simulate the attenuation bias, both these simulated variables $V_{a,\rho,k}$ and some real-world example exposures are then merged to the individual level data to construct (i) the observed exposures $X_{s,i}^*$ and (ii) the “true” exposures $X_{s,i}$.

The observed exposures $X_{s,i}^*$ in our simulations are assigned to UKB participants based on their reported birth date and birth location. The “true” exposures $X_{s,i}$ are assigned based on their reported birth date and:

- their birth location (if simulated to not have an error),
- their sibling’s birth location (if only this sibling is simulated to have an error),
OR
- the midpoint between the two siblings’ birth locations (if both siblings are simulated to have an error).

We have now clarified this approach in the “Methods” section, saying:

“In our simulations of the attenuation bias, the simulated time-varying spatially correlated variables $V_{a,\rho,k}$ as well as a variety of standardized district-level measures of disease exposure and demographics are merged with the sibling sample used in our main estimations to construct for each simulation run (i) the observed exposures $X_{s,i}^*$ and (ii) the “true” exposures $X_{s,i}$.

We begin by merging the time-varying area-level variables to the sibling sample based on each individual’s reported birth date and birth location; the resulting variable is defined as $X_{t,ownloc}$, where *ownloc* indicates the individual’s reported birth location. Additionally we merge these variables based on each individual’s reported birth date and their *sibling’s* reported birth location ($X_{t,sibloc}$), as well as the geographic midpoint between the two birth locations reported by the two siblings ($X_{t,midloc}$). If the geographic midpoint between the two birth locations is not on land, the closest UK land location to the midpoint is used.” (p.20)

[...]

“The “true” exposure in our simulations is defined as follows: [...] Irrespective of any errors to the reported birth location, the exposure $X_{s,i}$ is always defined based on the individual’s reported date of birth t . If only one sibling in a sibling pair is simulated to have an incorrect birth location, their sibling’s reported birth location *sibloc* is used to compute the exposure. If both siblings in a sibling pair are simulated to have an incorrect birth location, we use the mid-point *midloc* between their reported birth locations. In the absence of any information on the true birth location in these cases, the midpoint between the two reported locations sets a lower bound on the average geographic distance between the true and the observed birth locations. For all individuals who are not simulated to have an incorrect birth location, we use their own reported birth location *ownloc*.

The observed exposure in our simulations is defined as

$$X_{s,i}^* = X_{t,ownloc} \quad (8)$$

for all individuals and thus subject to measurement error due to misreported birth locations.” (p.21)

- iii. What does rho capture, exactly, in plain English? I understand that it’s the spatial autocorrelation parameter, but how is it scaled? For instance, what does a rho of 0.05 vs. 0.25 imply, exactly? Is it Moran’s I, or what metric is it based on?

Rho is the spatial autocorrelation (or autoregression) parameter in a spatial autoregressive model (SAR) that ranges from -1 to 1 (for a row-standardized spatial weighting matrix). It is a parameter specific to the SAR model used to generate the spatial data in our simulations, whereas Moran’s I measures spatial correlation in a more general way. We have modified the relevant parts of the “Methods” section to clarify this (p.19):

“We begin by simulating time-invariant spatially correlated data at the district, parish and coordinate level based on a spatial autoregressive model with spatial autocorrelation parameter ρ that ranges from -1 to 1 (if the spatial weighting matrix is row-standardized). A positive ρ corresponds to spatial clustering, with larger values of ρ indicating stronger spatial clustering. A negative ρ indicates spatial dispersion and ρ equals zero when there is no spatial autocorrelation. We use the `sim_sar` command of the `geostan` package (for parish- and district-level data) and the `powerWeights` command of the `spatialreg` package (for coordinate-level data) in R to simulate 10 variables $S_{a,\rho}$ for each spatial aggregation level a (coordinate-, parish- and district-level) and spatial autocorrelation parameter $\rho \in [0.00, 0.05, 0.10, \dots, 0.90, 0.95, 0.975]$.”

- iv. In formula (12), epsilon is standard normal, but that does not mean much unless we know what’s the scale (or the variance) of X? Does that vary across X’s, or do you standardize X? If the former, then it’s hard to compare the results in Figure 3.

Thank you for highlighting this. All the X variables (both the simulation-based and example exposures) have been standardized to have a mean of zero and a standard deviation of one. We have clarified this in the “Results” section (p.7) where we discuss the Figure as well as in the “Methods” section (p.19 and 21):

“We standardize the resulting time-varying spatially correlated $V_{a,\rho,k}$ to have a mean of zero and a standard deviation of one.”

“The exposures $X_{s,t}$ in our simulations are standardized to have a mean of zero and a standard deviation of one.”

REVIEWERS' COMMENTS

Reviewer #1 (Remarks to the Author):

Very comprehensive revision, all my comments have been dealt with fully.

Reviewer #1 (Remarks on code availability):

I have confirmed the GitHub exists and in fact contains code

Reviewer #2 (Remarks to the Author):

von Hinke and Vitt have satisfactorily responded to my main comments and have nicely updated their paper. This will be a valuable contribution to the literature.

This is a signed review: Jonathan P. Beauchamp